# Label Robust and Differentially Private Linear Regression: Computational and Statistical Efficiency

**Xiyang Liu**
Paul Allen School of Computer Science & Engineering
University of Washington
xiyangl@cs.washington.edu

**Prateek Jain**
Google Research
prajain@google.com

**Weihao Kong**
Google Research
weihaokong@google.com

**Sewoong Oh**
Paul Allen School of Computer Science & Engineering
University of Washington, and Google Research
sewoong@cs.washington.edu

**Arun Sai Suggala**
Google Research
arunss@google.com

## Abstract

We study the canonical problem of linear regression under $(\varepsilon, \delta)$-differential privacy when the datapoints are sampled i.i.d. from a distribution and a fraction of response variables are adversarially corrupted. We provide the first provably efficient – both computationally and statistically – method for this problem, assuming standard assumptions on the data distribution. Our algorithm is a variant of the popular differentially private stochastic gradient descent (DP-SGD) algorithm with two key innovations: a full-batch gradient descent to improve sample complexity and a novel adaptive clipping to guarantee robustness. Our method requires only linear time in input size, and still matches the information theoretical optimal sample complexity up to a data distribution dependent condition number factor. Interestingly, the same algorithm, when applied to a setting where there is no adversarial corruption, still improves upon the existing state-of-the-art and achieves a near optimal sample complexity.

## 1 Introduction

Differential Privacy (DP) [33] is a standard notion of privacy widely adopted by both industry and government [76, 35, 36, 2]. With widespread usage of ML and statistical techniques, DP becomes even more critical to ensure private information of participating individuals is not revealed in any form via the learned model. An statistical estimator is said to be $(\varepsilon, \delta)$-differentially private if presence/absence of an individual's data point in the dataset does not significantly change the estimated output. Smaller $\varepsilon > 0$ and $\delta \in [0, 1]$ imply stronger privacy guarantees.

While privacy preserving statistical estimators have been studied extensively in recent past, several critical questions remain open (see App. A for a survey). Consider the canonical statistical task of linear regression with $n$ i.i.d. samples, $\{(x_i \in \mathbb{R}^d, y_i \in \mathbb{R})\}_{i=1}^n$, drawn from $x_i \sim \mathcal{N}(0, \Sigma)$, $y_i = x_i^\top w^* + z_i$, $z_i \sim \mathcal{N}(0, \sigma^2)$ and $\mathbb{E}[x_i z_i] = 0$ for some true parameter $w^* \in \mathbb{R}^d$. The error is measured in $(1/\sigma)\|\hat{w} - w^*\|_\Sigma := (1/\sigma)\|\Sigma^{1/2}(\hat{w} - w^*)\|$, which correctly accounts for the signal-to-noise ratio in each direction; in the direction of large eigenvalue of $\Sigma$, we have larger signal in $x_i$

37th Conference on Neural Information Processing Systems (NeurIPS 2023).

but the noise $z_i$ remains the same. We expect smaller errors in those directions, which is accounted for in the error measure $(1/\sigma)\|\hat{w} - w^*\|_\Sigma$.

Minimax optimal sample complexity for estimating the optimal linear regression model with DP was recently established. For the lower bound, using recently introduced score attack technique, [16, Theorem 3.1] shows that $n = \Omega(d/\alpha^2 + d/(\varepsilon\alpha))$ samples are necessary to achieve an error of $(1/\sigma)\|\hat{w} - w^*\|_\Sigma = \alpha$ (in expectation). For the matching upper bound, High-dimensional Propose-Test-Release (HPTR) in [61] and Robust-to-Private in [10] show that $n = \tilde{O}(d/\alpha^2 + d/(\varepsilon\alpha))$ samples are also sufficient. The first term of $d/\alpha^2$ is the fundamental sample complexity even if privacy is not required, and the second term of $d/(\varepsilon\alpha)$ is the cost of privacy.

This implies that, statistically, the problem appears to be solved. However, computationally, the problem is still open despite multiple studies of the problem. That is, the statistical optimal algorithms still take exponential time.

After a series of efforts in computationally efficient approaches as surveyed in App. A, [81] achieves the best known sample complexity of $n = \tilde{O}(d/\alpha^2 + \kappa d/(\varepsilon\alpha) + \kappa^2 d/\varepsilon)$, where $\kappa$ is the condition number of the covariance $\Sigma$ of the covariates. Compared to HPTR, the cost of computational efficiency is factor of $\kappa$ in the second term and the third term that is unnecessary. As the condition number can be quite large, improving the dependence on $\kappa$ is of utmost importance. Furthermore, the technique of [81] strictly requires sampling without replacement, whose analysis relies on having an explicit form of the end-to-end update. In particular, their analysis technique is not applicable to the case with corrupted samples.

In contrast, we propose a novel method (Alg. 1) that builds upon full-batch gradient descent and applies a carefully chosen adaptive clipping which is a general technique used in practice as well [1]. Together with an intuitive but intricate analysis technique, we improve the sample complexity to $n = \tilde{O}(d/\alpha^2 + \kappa^{1/2}d/(\varepsilon\alpha))$.

**Corollary 1.1** (Corollary of Thm. 3.1 for sub-Gaussian data). *Alg. 1 is $(\varepsilon, \delta)$-DP. Let $S = \{(x_i, y_i)\}_{i=1}^n$ be a dataset of i.i.d. samples with $x_i \sim \mathcal{N}(0, \Sigma)$, $y_i = x_i^\top w^* + z_i$ and $z_i \sim \mathcal{N}(0, \sigma^2)$ for some unknown true parameter $w^* = \Sigma^{-1}\mathbb{E}[y_i x_i] \in \mathbb{R}^d$ and unknown $\Sigma$ and $\sigma^2$. Then $n = \tilde{O}(d/\alpha^2 + \kappa^{1/2}d/(\varepsilon\alpha))$ samples are sufficient for Alg. 1 to achieve $(1/\sigma)\|\hat{w} - w^*\|_\Sigma = \tilde{O}(\alpha)$ with high probability, where $\kappa := \lambda_{\max}(\Sigma)/\lambda_{\min}(\Sigma)$.*

Due to space constraints, we focus on sub-Gaussian distributions in the main text and provide comparisons to prior work in Tab. 1. Our analysis in App. H applies to a more general family of *light-tailed* distributions, called sub-Weibull. Next, when the noise in the samples is *heavy-tailed*, a similar algorithm can be applied with carefully chosen clipping thresholds to account for the heavier tail. Concretely, for $k$-th moment bounded distributions, the tail of the distribution gets increasingly heavier with smaller $k$. This would require larger number of samples to achieve the same accuracy, which is captured in our sample complexity of $n = \tilde{O}(d/\alpha^{2k/(k-1)} + \kappa^{1/2}d/(\varepsilon\alpha^{k/(k-1)}))$. We explain the heavy-tailed setting, provide a detailed analysis and a proof, and discuss the results in App. L. This is the first efficient algorithm with provable guarantees achieving $(\varepsilon, \delta)$-DP.

**Corollary 1.2** (informal version of Coro. L.7 for heavy-tailed noise). *Alg. 4 is $(\varepsilon, \delta)$-DP. Let $S = \{(x_i, y_i)\}_{i=1}^n$ be a dataset of i.i.d. samples with $x_i \sim \mathcal{N}(0, \Sigma)$, $y_i = x_i^\top w^* + z_i$, and the zero-mean, independent, and heavy-tailed noise $z_i$ satisfies $\mathbb{E}[|z/\sigma|^k] = O(1)$ for some unknown true parameter $w^* \in \mathbb{R}^d$ and unknown $\Sigma$ and $\sigma^2$. Then $n = \tilde{O}(d/\alpha^{2k/(k-1)} + \kappa^{1/2}d/(\varepsilon\alpha^{k/(k-1)}))$ samples are sufficient for Alg. 4 in App. L to achieve an error rate of $(1/\sigma)\|\hat{w} - w^*\|_\Sigma = \tilde{O}(\alpha)$ with high probability, where $\kappa := \lambda_{\max}(\Sigma)/\lambda_{\min}(\Sigma)$.*

Perhaps surprisingly, we show that Alg. 1 is also robust against label-corruption, where an adversary selects an arbitrary $\alpha_{\text{corrupt}}$ fraction of the data points and changes their response variables arbitrarily. Ideally, we want a robust algorithm against a stronger adversary who can corrupt the covariates also. However, even for a simpler problem of private mean estimation, achieving robustness against such a strong adversary with $O(d)$ samples requires heavy machinery (convex relaxations of sum-of-squares optimization) with significantly more computations (although polynomial) [45].

Our lower bound in Prop. 3.9, together with the lower bound in [16] on the uncorrupted case, shows that $n = \Omega(d/\alpha^2 + d/(\varepsilon\alpha))$ samples are necessary to achieve an error rate of $(1/\sigma)\|\hat{w} - w^*\|_\Sigma = O(\alpha + \alpha_{\text{corrupt}})$. In particular, it is impossible to achieve an error below $\alpha_{\text{corrupt}}$ even if we have

Table 1: Suppose data is drawn from a linear model in $d$-dimensions from sub-Gaussian covariates with covariance $\Sigma$ and sub-Gaussian noise with variance $\sigma^2$. To achieve an error rate of $(1/\sigma)\|\hat{w} - w^*\|_\Sigma = \alpha$ with $(\varepsilon, \delta)$-DP, DP-RobGD requires the least number of samples among computationally efficient algorithms. This improves over [81] by a factor of $\kappa^{1/2}$ in the second term, where $\kappa$ is the condition number of $\Sigma$. We hide polylogarithmic factors in $d$, $\kappa$ and $1/\delta$. $\spadesuit$DP-Theil-Sen is only analyzed when $\kappa = 1$ and its dependence $\kappa^c$ is unknown.

| Algorithm | Runtime | Sample Complexity |
|---|---|---|
| TukeyEM [6] | poly | no guarantee |
| DP-Theil-Sen [74] $\spadesuit$ | poly | $\frac{d^2}{\alpha^2} + \frac{d}{\varepsilon\alpha}\kappa^c$ |
| DP-AMBSSGD [81] | poly | $\frac{d}{\alpha^2} + \frac{d}{\varepsilon\alpha}\kappa + \frac{\kappa^2 d}{\varepsilon}$ |
| DP-RobGD [Theorem 3.8] | poly | $\frac{d}{\alpha^2} + \frac{d}{\varepsilon\alpha}\kappa^{1/2}$ |
| HPTR [61], Robust-to-private [10] | exp | $\frac{d}{\alpha^2} + \frac{d}{\varepsilon\alpha}$ |
| Lower Bound [16] | | $\frac{d}{\alpha^2} + \frac{d}{\varepsilon\alpha}$ |

infinite samples (Prop. 3.9), and hence there is no need to aim for $\alpha < \alpha_{\text{corrupt}}$. This lower bound is matched by exponential time approaches, HPTR in [61] and Robust-to-Private in [10], which also guarantee robustness. Currently, there is no efficient algorithm that can guarantee both privacy and robustness for linear regression. To this end, we provide the first efficient algorithm guaranteeing both, with a sample complexity that is optimal up to a $\kappa^{1/2}$ factor.

**Corollary 1.3** (Corollary of Thm. H.3 for sub-Gaussian data with adversarial label corruption). *Under the hypotheses of Coro. 1.1, suppose $\alpha_{\text{corrupt}}$-fraction of the labels are corrupted arbitrarily. Then $n = \tilde{O}(d/\alpha^2 + \kappa^{1/2}d/(\varepsilon\alpha))$ samples are sufficient for Alg. 1 to achieve an error rate of $(1/\sigma)\|\hat{w} - w^*\|_\Sigma = \tilde{O}(\alpha + \alpha_{\text{corrupt}})$ with high probability, where $\kappa := \lambda_{\max}(\Sigma)/\lambda_{\min}(\Sigma)$.*

When $\alpha_{\text{corrupt}} = 0$, this recovers the non-robust result from Coro. 1.1. A similar robustness guarantee also holds for heavy-tailed settings. We provide a formal statement in App. L

**Contributions.** For a canonical problem of private linear regression under sub-Gaussian distributions, the best known efficient algorithm [81] requires

$$n = \tilde{O}\left(\frac{d}{\alpha^2} + \frac{\kappa d}{\varepsilon\alpha} + \frac{\kappa^2 d}{\varepsilon}\right),$$

to achieve $(1/\sigma)\|\hat{w} - w^*\|_\Sigma = \alpha$. We provide the first efficient algorithm that improves this to

$$n = \tilde{O}\left(\frac{d}{\alpha^2} + \frac{\kappa^{1/2} d}{\varepsilon\alpha}\right),$$

which nearly matches the exponential-time algorithms [61, 10] and the lower bound [16] up to $\kappa^{1/2}$ in the second term. For the same problem, we show that the same algorithm is the first to achieve robustness against adversarial corruption of the labels.

Under a heavy-tailed distribution of the noise, we provide the first computationally efficient algorithm, to the best of our knowledge, that achieves a sample complexity close to that of an exponential-time algorithm of [61]. There is no matching lower bound in the heavy-tailed setting. This is also the first efficient algorithm to achieve robustness against adversarial corruption of the labels under heavy-tailed noise.

## 2 Problem formulation and background

When there is no adversary, we present our results under the standard linear model with sub-Gaussian covariates and noise. In App. H, we present a more general family of $(K, a)$-sub-Weibull distributions that recovers the standard sub-Gaussian family as a special case when $a = 0.5$. The necessity of such assumptions on the tail is explained in Sec. 3.4.

**Assumption 2.1** (sub-Gaussian model). We have i.i.d. samples $S = \{(x_i \in \mathbb{R}^d, y_i \in \mathbb{R})\}_{i=1}^n$ from a distribution $\mathcal{P}_{\Sigma, w^*, \sigma^2}$ of a linear model $y_i = \langle x_i, w^* \rangle + z_i$, where the input vector $x_i$ has zero

mean $\mathbb{E}[x_i] = 0$ and a positive definite covariance $\Sigma := \mathbb{E}[x_i x_i^\top] \succ 0$, and the (input dependent) label noise $z_i$ has zero mean $\mathbb{E}[z_i] = 0$ and variance $\sigma^2 := \mathbb{E}[z_i^2]$. We further assume $\mathbb{E}[x_i z_i] = 0$, which is equivalent to assuming that the true parameter $w^* = \Sigma^{-1}\mathbb{E}[y_i x_i]$. We assume the marginal distributions of $x_i$ and $z_i$ are $K$-sub-Gaussian with $K = O(1)$, as defined below.

**Definition 2.2.** $x \in \mathbb{R}^d$ is $K$-sub-Gaussian if for all $v \in \mathbb{R}^d$, $\mathbb{E}\left[\exp\left(\frac{\langle v,x\rangle^2}{K^2\mathbb{E}[\langle v,x\rangle^2]}\right)\right] \leq 2$.

Given a dataset $S$ that is i.i.d. sampled from $\mathcal{P}_{\Sigma^2,w^*,\sigma^2}$ satisfying Asmp. 2.1, our goal is to estimate $w^*$ that minimizes $(1/\sigma)\|\hat{w} - w^*\|_\Sigma$ which is also equivalent to minimize the excess population risk, i.e., $\mathcal{L}(w^*) - \mathcal{L}(\hat{w})$ where $\mathcal{L}(w) := \mathbb{E}_{(x,y)\sim\mathcal{P}_{\Sigma,w^*,\sigma^2}}[(y - \langle w,x\rangle)^2]$.

**Notations.** A vector $x \in \mathbb{R}^d$ has the Euclidean norm $\|x\|$. For a matrix $M$, we use $\|M\|_2$ to denote the spectral norm. The error is measured in $\|\hat{w} - w^*\|_\Sigma := \|\Sigma^{1/2}(\hat{w} - w^*)\|$ for some PSD matrix $\Sigma$. The identity matrix is denoted by $\mathbf{I}_d \in \mathbb{R}^{d\times d}$. Let $[n] = \{1, 2, \ldots, n\}$. $\tilde{O}(\cdot)$ hides some constants terms, $K = \Theta(1)$, and poly-logarithmic terms in $n$, $d$, $1/\varepsilon$, $\log(1/\delta)$, $1/\zeta$, and $1/\alpha_{\text{corrupt}}$. For a vector $x \in \mathbb{R}^d$, we define $\text{clip}_a(x) := x \cdot \min\{1, a/\|x\|\}$.

**Background on DP.** Differential Privacy is a standard measure of privacy leakage when data is accessed via queries, introduced by [33]. Two datasets $S$ and $S'$ are said to be neighbors if they differ at most by one entry, which is denoted by $S \sim S'$. A stochastic query $q$ is said to be $(\varepsilon, \delta)$-differentially private for some $\varepsilon > 0$ and $\delta \in [0, 1]$, if $\mathbb{P}(q(S) \in A) \leq e^\varepsilon\mathbb{P}(q(S) \in A) + \delta$, for all neighboring datasets $S \sim S'$ and all subset $A$ of the range of the query. We build upon two widely used DP primitives, the Gaussian mechanism and the private histogram. A central concept in DP mechanism design is the *sensitivity* of a query, defined as $\Delta_q := \sup_{S\sim S'}\|q(S) - q(S')\|$. We describe Gaussian mechanism and private histogram in App. B.

## 2.1 Comparisons with the prior work

The state-of-the-art approach introduced by [81] is based on DP-SGD [71], where privacy is ensured by gradient norm clipping and the Gaussian mechanism. Two additional technical components are adaptive clipping and streaming SGD. Adaptive clipping with an appropriate threshold $\theta_t$ ensures that no data point is clipped (under the sub-Gaussian assumption), while providing a bound on the sensitivity of the average mini-batch gradient (to ensure we do not add too much noise). The streaming approach, where each data point is only touched once and discarded, ensures independence between the past iterate $w_t$ and the gradients at round $t + 1$, which the analysis critically relies on. For $T = \tilde{\Theta}(\kappa)$ iterations where $\kappa$ is the condition number of the covariance $\Sigma$, the dataset $S = \{(x_i, y_i)\}_{i=1}^n$ is partitioned into $\{B_t\}_{t=0}^{T-1}$ subsets of equal size: $|B_t| = \tilde{\Theta}(n/\kappa)$. At each round $t$, the gradients are clipped and averaged with additive Gaussian noise chosen to satisfy $(\varepsilon, \delta)$-DP:

$$w_{t+1} \leftarrow w_t - \eta\left(\frac{1}{|B_t|}\sum_{i\in B_t}\text{clip}_{\theta_t}(x_i(w_t^\top x_i - y_i)) + \frac{\theta_t\sqrt{2\log(1.25/\delta)}}{\varepsilon|B_t|}\nu_t\right), \quad (1)$$

where $\nu_t \sim \mathcal{N}(0, \mathbf{I}_d)$. In [81], a slight variation of this streaming SGD is shown to achieve an error of $(1/\sigma)\|w_T - w^*\|_\Sigma = \alpha$ with $n = \tilde{O}(d/\alpha^2 + \kappa d/(\varepsilon\alpha) + \kappa^2 d/\varepsilon)$ samples (Row 3 in Tab. 1).

**Our technical innovations.** Our approach builds upon such gradient based methods but makes several important innovations. First, we use full-batch gradient descent, as opposed to the streaming SGD above. Using all $n$ samples reduces the sensitivity of the per-round gradient average by a $\kappa$ factor, and thus decreases the privacy noise added in each iteration. This improves the second term of sample complexity from $\kappa d/(\varepsilon\alpha)$ to $\kappa^{1/2}d/(\varepsilon\alpha)$ and removes the third term completely. However, full-batch GD loses the independence that the streaming SGD enjoyed between $w_t$ and the samples used in the round $t + 1$. This dependence makes the analysis more challenging. We instead propose using the *resilience* to precisely track the bias and variance of the (dependent) full-batch average gradient. Resilience is a central concept in robust statistics that links the tail-property of the distribution to the bias, which we explain in Sec. 5.

Next, one critical component in achieving this improved sample complexity is the new analysis technique we introduce for tracking the end-to-end gradient updates. Since our gradient descent algorithm is not guaranteed to make progress every step, we cannot use the vanilla one-step analysis.

Taking the full end-to-end analysis by expanding the whole gradient trajectory will introduce too many correlated cross-terms which are very hard to control. Therefore, we leverage an every $\kappa$-step analysis and show that the objective function at least decreases geometrically every $\kappa$ steps. To be more specific, our analysis technique in App. H (steps 3 and 4) opens up the iterative updates from the beginning to the end, and exploits the fact that $\lambda_{\max}((\eta\Sigma)^{1/2}(1 - \eta\Sigma)^i(\eta\Sigma)^{1/2})$ is upper bounded by $1/(i + 1)$ when $\|\eta\Sigma\| \leq 1$. This technique is critical in achieving the near-optimal dependence in $\kappa$. This might be of independent interest to other analysis of gradient-based algorithms. We refer to the beginning of step 3 in App. H for a detailed explanation.

Finally, we propose a novel clipping that separately clips $x_i$ and $(w_t^\top x_i - y_i)$ in the gradient, $(w_t^\top x_i - y_i)x_i$. This is critical in achieving robustness to label-corruption, as we explain in Sec. 3.1.

## 3 Label-robust and private linear regression

We introduce a novel gradient descent approach. This achieves an improved sample complexity compared to the state-of-the-art algorithm and robustness against label corruption.

### 3.1 Algorithm

The skeleton of our approach in Alg. 1 is the general DP-SGD [1, 71] with adaptive clipping [7]. We partition the dataset into three equal-sized subsets: $S_1, S_2, S_3$. $S_1$ and $S_2$ are used in adaptively estimating the clipping thresholds, and $S_3$ is re-used every step to compute the average gradient.

The standard adaptive clipping, e.g., [7, 81], is not robust against label-corruption. Under sub-Gaussian distribution, a positive fraction of the covariates, $x_i$'s, can be close to the origin. If the adversary chooses to corrupt those points with small norm, $\|x_i\|$, they can make large changes in the corrupted residual, $(y_i - w_t^\top x_i)$, while evading the standard clipping by the norm of the gradient; the norm of the gradient, $\|x_i(y_i - w_t^\top x_i)\| = \|x_i\| \, |y_i - w_t^\top x_i|$, can remain under the threshold. This is problematic, since the bias due to the corrupted samples in the gradient scales proportionally to the magnitude of the residual (after clipping). To this end, we propose clipping the norm and the residual separately: $\mathrm{clip}_\Theta(x_i)\mathrm{clip}_{\theta_t}\left(w_t^\top x_i - y_i\right)$. This keeps the sensitivity of gradient average bounded by $\Theta(\theta_t)$. The subsequent Gaussian mechanism in line 11 ensures $(\varepsilon_0, \delta_0)$-DP at each round. Applying advanced composition in Lemma B.5 of $T$ rounds, this ensures end-to-end $(\varepsilon, \delta)$-DP.

**Novel adaptive clipping.** When clipping with $\mathrm{clip}_\Theta(x_i)$, the only purpose of clipping the covariate by its norm, $\|x_i\|$, is to bound the sensitivity of the resulting clipped gradient. In particular, we do not need to make it robust as there is no corruption in the covariates. Ideally, we want to select the smallest threshold $\Theta$ that does not clip any of the covariates. Since the norm of a covariate is upper bounded by $\|x_i\|^2 \leq K^2\mathrm{Tr}(\Sigma)\log(1/\zeta)$ with probability $1 - \zeta$ (Lemma J.3), we estimate the unknown $\mathrm{Tr}(\Sigma)$ using Private Norm Estimator in Alg. 3 in App. F and set the norm threshold $\Theta = K\sqrt{2\Gamma\log(n/\zeta)}$ (Alg. 1 line 4). The $n$ in the logarithm ensures that the union bound holds.

When clipping with $\mathrm{clip}_{\theta_t}(w_t^\top x_i - y_i)$, the purpose of clipping the residual by its magnitude, $|y_i - w_t^\top x_i| = |(w^* - w_t)^\top x_i + z_i|$, is to bound the sensitivity of the gradient and also to provide robustness against label-corruption. We want to choose a threshold that only clips corrupt data points and at most a few clean data points. In order to achieve an error $(1/\sigma)\|w_T - w^*\|_\Sigma = \alpha$, we know that any set of $(1 - \alpha)$ fraction of the clean data points is sufficient to get a good estimate of the average gradient. By clipping at $|(w^* - w_t)^\top x_i + z_i|^2 \leq (\|w_t - w^*\|_\Sigma^2 + \sigma^2)CK^2\log(1/(2\alpha))$, Lemma J.3 guarantees that the unclipped subset will be large enough, i.e., $(1 - \alpha)n$. At the same time, this threshold on the residual is small enough to guarantee robustness against the label-corrupted samples. We introduce the robust and DP Distance Estimator in Alg. 2 to estimate the unknown (squared and shifted) distance, $\|w_t - w^*\|_\Sigma^2 + \sigma^2$, and set the distance threshold $\theta_t = 2\sqrt{2\gamma_t}\sqrt{9C_2K^2\log(1/(2\alpha))}$ (Alg. 1 line 7). Both norm and distance estimation rely on DP histogram (Lemma B.2), but over a set of statistics computed on partitioned datasets, which we explain in detail in App. C.

---

**Algorithm 1:** Robust and Private Linear Regression

---

**Input:** $S = \{(x_i, y_i)\}_{i=1}^{3n}$, DP parameters $(\varepsilon, \delta)$, $T$, learning rate $\eta$, failure probability $\zeta$, target error $\alpha$, distribution parameter $K$

1   Partition dataset $S$ into three equal sized disjoint subsets $S = S_1 \cup S_2 \cup S_3$.

2   $\delta_0 \leftarrow \frac{\delta}{2T}, \varepsilon_0 \leftarrow \frac{\varepsilon}{4\sqrt{T \log(1/\delta_0)}}, \zeta_0 \leftarrow \frac{\zeta}{3}, w_0 \leftarrow 0$

3   $\Gamma \leftarrow \text{PrivateNormEstimator}(S_1, \varepsilon_0, \delta_0, \zeta_0)$        `// using Alg. 3, App. F`

4   $\Theta \leftarrow K\sqrt{2\Gamma} \log^a(n/\zeta_0)$

5   **for** $t = 0, 1, 2, \ldots, T-1$ **do**

6      $\gamma_t \leftarrow \text{PrivateDistanceEstimator}(S_2, w_t, \varepsilon_0, \delta_0, \alpha, \zeta_0)$    `// using Alg. 2, App. C`

7      $\theta_t \leftarrow 2\sqrt{2\gamma_t} \cdot \sqrt{9 C_2 K^2 \log(1/(2\alpha))}$.

8      Sample $\nu_t \sim \mathcal{N}(0, \mathbf{I}_d)$

9      $\tilde{g}_i^{(t)} \leftarrow \text{clip}_\Theta(x_i)\text{clip}_{\theta_t}(x_i^\top w_t - y_i)$

10     $\phi_t = (\sqrt{2\log(1.25/\delta_0)}\Theta\theta_t)/(\varepsilon_0 n)$

11     $w_{t+1} \leftarrow w_t - \eta\left(\frac{1}{n}\sum_{i \in S_3} \tilde{g}_i^{(t)} + \phi_t \nu_t\right)$

12   Return $w_T$

---

### 3.2   Analysis without adversarial corruption

We show that Alg. 1 achieves an improved sample complexity. We provide the proof for a more general class of distributions in App. H and a sketch of the proof in Sec. 5. We address the necessity of the assumptions in Sec. 3.4, along with some lower bounds.

**Theorem 3.1.** *Alg. 1 is $(\varepsilon, \delta)$-DP. Under sub-Gaussian model of Asmp. 2.1, for any failure probability $\zeta \in (0, 1)$ and target error rate $\alpha$, if the sample size is large enough such that*

$$n = \tilde{O}\left(K^2 d \log^2\left(\frac{1}{\zeta}\right) + \frac{d + \log(1/\zeta)}{\alpha^2} + \frac{K^2 d T^{1/2} \log(\frac{1}{\delta})\sqrt{\log(\frac{1}{\zeta})}}{\varepsilon\alpha}\right), \tag{2}$$

*with a large enough constant, then the choices of a step size $\eta = 1/(C\lambda_{\max}(\Sigma))$ for any $C \geq 1.1$ and the number of iterations, $T = \tilde{\Theta}(\kappa \log(\|w^*\|))$ for a condition number of the covariance $\kappa := \lambda_{\max}(\Sigma)/\lambda_{\min}(\Sigma)$, ensures that, with probability $1 - \zeta$, Alg. 1 achieves*

$$\mathbb{E}_{\nu_1, \cdots, \nu_T \sim \mathcal{N}(0, \mathbf{I}_d)}\left[\|w_T - w^*\|_\Sigma^2\right] = \tilde{O}\left(K^4 \sigma^2 \alpha^2 \log^2\left(\frac{1}{\alpha}\right)\right), \tag{3}$$

*where the expectation is taken over the noise added for DP, and $\tilde{O}$ and $\tilde{\Theta}(\cdot)$ hide logarithmic terms in $K, \sigma, d, n, 1/\varepsilon, \log(1/\delta), 1/\alpha$, and $\kappa$.*

*Remark* 3.2. Omitting some constant and logarithmic terms, Alg. 1 requires

$$n = \tilde{O}\left(\frac{d}{\alpha^2} + \frac{\kappa^{1/2} d}{\varepsilon\alpha}\right), \tag{4}$$

samples to ensure an error rate of $(1/\sigma^2)\mathbb{E}[\|w_T - w^*\|_\Sigma^2] = \tilde{O}(\alpha^2)$. From [16, Theorem 3.1], there exists an $n = \Omega(d/\alpha^2 + d/(\varepsilon\alpha))$ lower bound, and our upper bound matches this lower bound up to a factor of $\kappa^{1/2}$ in the second term and other logarithmic factors. Eq. (4) is the best known rate among all efficient private linear regression algorithms, strictly improving upon the state-of-the-art. The best existing efficient algorithm by [81] requires $n = \tilde{O}(d/\alpha^2 + \kappa d/(\varepsilon\alpha) + \kappa^2 d/\varepsilon)$ to achieve the same error rate. Compared to Eq. (4), the second term is larger by a factor of $\kappa^{1/2}$ compared to the second term in Eq. (4). Further, [81] requires $\kappa^2 d/\varepsilon$, which is not needed in Eq. (4).

*Remark* 3.3. Consider the standard settings of linear regression with $x_i \sim \mathcal{N}(0, \mathbf{I}_d)$ and $z_i \sim \mathcal{N}(0, \sigma^2)$ such that the condition number is one, our bound given by Eq. (4) nearly matches the lower bound ([16, Theorem 3.1]) up to logarithmic factors.

*Remark* 3.4. Note that the leading term in Eq. (4) is the first term $d/\alpha^2$ when target error $\alpha \leq \varepsilon/\kappa^{1/2}$. Our first term is independent of $\kappa$, which matches the lower bound for non-private linear regression.

*Remark* 3.5. The third term $\kappa^2 d/\varepsilon$ in [81] is independent of error rate $\alpha$ but scales as $\kappa^2$. This term is required to ensure the privacy noise added in each iteration is small enough for their DP-SGD to make progress (Appendix. B.2.2 in [81]). Our algorithm is based on full-batch gradient descent, which uses all $n$ samples and thus reduces the sensitivity of gradient average by a $\kappa$ factor. As a result, we show in Eq. (59) that our algorithm only requires $n = \tilde{O}((1/\varepsilon)\sqrt{\kappa^{1/2}d/\alpha})$ to make progress for each iteration. This is strictly smaller than our dominant term $\kappa^{1/2}d/(\varepsilon\alpha)$ and does not show up in our final guarantee. We provide a formal proof in App. H.

*Remark* 3.6. One of the key innovations in Alg. 1 is the adaptive distance estimator (Alg. 2 in App. C). The goal is to privately estimate the (shifted) distance of the current estimate, i.e., $\|w_t - w^*\|_\Sigma + \sigma^2$, without the knowledge of $w^*$. We show in Thm. C.1 that our novel distance estimator only requires an error-independent sample complexity $n = \tilde{O}(\kappa^{1/2}d/\varepsilon)$ to achieve a constant multiplicative error. Note that the DP-STAT (Algorithm 3 in [81]) can also be used to estimate the distance. But it requires the knowledge of domain size $\|w^*\|_\Sigma + \sigma$. We completely remove this requirement, improve the dependence on $K$ and $\log(n)$, and show it is also robust, as introduced in the next section. We provide the algorithms and analysis in App. C and the formal proof in App. D.

### 3.3 Robustness against label corruption

We assume there exists a good dataset $S_{\text{good}}$ that satisfies Asmp. 2.1. We only get access to a label-corrupted dataset under the standard definition of *label corruption*, e.g., [15]. There are variations in literature on the definition, which we survey in App. A.

**Assumption 3.7** ($\alpha_{\text{corrupt}}$-corruption). Given a dataset $S_{\text{good}} = \{(x_i, y_i)\}_{i=1}^n$, an adversary inspects all the data points, selects $\alpha_{\text{corrupt}}n$ data points denoted as $S_r$, and replaces the labels with arbitrary labels while keeping the covariates unchanged. We let $S_{\text{bad}}$ denote this set of $\alpha_{\text{corrupt}}n$ newly labelled examples by the adversary. Let the resulting set be $S_{\text{corrupt}} := S_{\text{good}} \cup S_{\text{bad}} \setminus S_r$.

Our goal is to estimate the unknown parameter $w^*$, given corrupted dataset $S_{\text{corrupt}}$, distribution parameter $K$, and (an upper bound on) the corruption level $\alpha_{\text{corrupt}}$.

Under the *non-private scenario*, i.e., $\varepsilon = \infty$, recent advances led to optimal algorithms for linear regression that are robust to label corruptions [15, 21]; if the corruption level is smaller than the target error rate, i.e., $\alpha_{\text{corrupt}} \leq \alpha$, then $n = \tilde{O}(d/\alpha^2)$ samples are sufficient to achieve an error rate of $(1/\sigma)\|\hat{w} - w^*\|_\Sigma = \alpha$. The sample complexity of $d/\alpha^2$ is optimal as it matches the information theoretic lower bound. The condition $\alpha_{\text{corrupt}} \leq \alpha$ is necessary since it is information theoretically impossible to achieve error $\alpha$ less than $\alpha_{\text{corrupt}}$, as we prove in Prop. 3.9. Setting the target error to the minimum possible value of $\alpha = \alpha_{\text{corrupt}}$, we say that these algorithms achieve optimal robustness since the minimum robust error rate of $(1/\sigma)\|\hat{w} - w^*\|_\Sigma = O(\alpha_{\text{corrupt}})$ can be achieved with minimal sample complexity of $n = \tilde{O}(d/\alpha_{\text{corrupt}}^2)$. We aim to achieve such optimal robustness simultaneously with differential privacy in a computationally efficient manner.

**Theorem 3.8.** *Under sub-Gaussian model of Asmp. 2.1 and $\alpha_{\text{corrupt}}$-corruption of Asmp. 3.7, if the corruption level is below the target error rate, $\alpha \geq \alpha_{\text{corrupt}}$, then $n = \tilde{O}(d/\alpha^2 + \kappa^{1/2}d/(\varepsilon\alpha))$ samples are sufficient for Alg. 1 to achieve an error rate of $(1/\sigma^2)\mathbb{E}[\|\hat{w} - w^*\|_\Sigma^2] = \tilde{O}(\alpha^2)$.*

This is the first efficient approach to achieve robustness and $(\varepsilon, \delta)$-DP simultaneously. The existing such algorithms take exponential time [61, Corollary C.2] and [10], but achieve optimal sample complexity of $n = O(d/\alpha^2 + d/(\varepsilon\alpha))$. Notice that there is no dependence on $\kappa$. It remains an open question if *computationally efficient* private linear regression algorithms can achieve such an optimal $\kappa$-independent sample complexity. We make the first advance towards this ambitious goal with the above theorem. Our sample complexity is sub-optimal only by a factor of $\sqrt{\kappa}$ in the second term. This is achieved by individually clipping the covariate, $x_i$, and the residual, $(w_t^\top x_i - y_i)$, in Alg. 1 and carefully tracking the bias of clipping with the use of resilience in the analysis in App. H.

### 3.4 Lower bounds

**Necessity of our assumptions.** A tail assumption on the covariate $x_i$ such as Asmp. 2.1 is necessary to achieve $n = O(d)$ sample complexity in Eq. (4). Even when the covariance $\Sigma$ is close to identity, without further assumptions on the tail of covariate $x$, the result in [13] implies that for $\delta < 1/n$, it is necessary for an $(\varepsilon, \delta)$-DP estimator to have $n = \Omega(d^{3/2}/(\varepsilon\alpha))$ samples to achieve

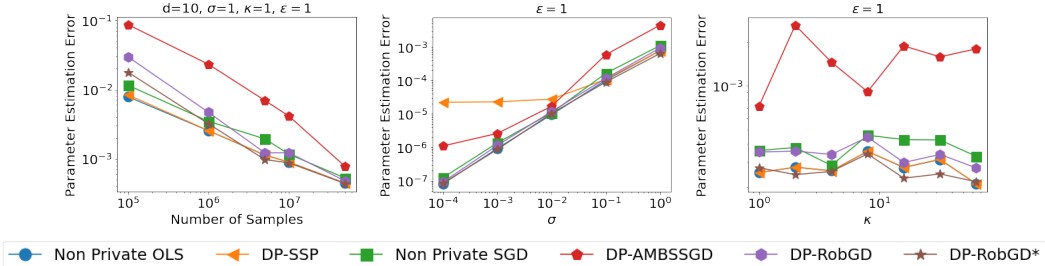

Figure 1: Performance of various techniques on DP linear regression. $d = 10$ in all the experiments. $n = 10^7, \kappa = 1$ in the $2^{nd}$ experiment. $n = 10^7, \sigma = 1$ in the $3^{rd}$ experiment, where $\kappa$ is the condition number of $\Sigma$ and $\sigma^2$ is the variance of the label noise $z_i$.

$\|\hat{w} - w^*\|_\Sigma = \tilde{O}(\alpha)$ (see Eq. (3) in [83]). Note that this lower bound is a factor $d^{1/2}$ larger than our upper bound that benefits from the additional tail assumption.

A tail assumption on the noise $z_i$ such as Asmp. 2.1 is necessary to achieve $n = O(d/(\varepsilon\alpha))$ dependence on the sample complexity in Eq. (4). For heavy-tailed noise, such as $k$-th moment bounded noise, the dependence can be significantly larger. [61, Proposition C.5] implies that for $\delta = e^{-\Theta(d)}$ and 4-th moment bounded $x_i$ and $z_i$, any $(\varepsilon, \delta)$-DP estimator requires $n = \Omega(d/(\varepsilon\alpha^2))$, which is a factor of $1/\alpha$ larger, to achieve $(1/\sigma^2)\|\hat{w} - w^*\|_\Sigma = \tilde{O}(\alpha)$.

The assumption that only labels are corrupted is critical for Alg. 1. The average of the clipped gradients can be significantly more biased, if the adversary can place the covariates of the corrupted samples in the same direction. In particular, the bound on the bias of our gradient step in Eq. (44) in App. H would no longer hold. Against such strong attacks, one requires additional steps to estimate the mean of the gradients robustly and privately, similar to those used in robust private mean estimation [60, 56, 44, 8]. There is no known linear-time algorithm to achieve this, and this is outside the scope of this paper.

**Lower bounds under label corruption.** Under the $\alpha_{\text{corrupt}}$ label corruption setting (Asmp. 3.7), even with infinite data and without privacy constraints, no algorithm is able to learn $w^*$ with $\ell_2$ error better than $\alpha_{\text{corrupt}}$. We provide a formal derivation for completeness.

**Proposition 3.9.** *Let $\mathcal{D}_{\Sigma,\sigma^2,w^*,K}$ be a class of distributions on $(x_i, y_i)$ from sub-Gaussian model in Asmp. 2.1. Let $S_{n,\alpha}$ be an $\alpha$-corrupted dataset of $n$ i.i.d. samples from some distribution $\mathcal{D} \in \mathcal{D}_{\Sigma,\sigma^2,w^*,K}$ under Asmp. 3.7. Let $\mathcal{M}$ be a class of estimators that are functions over $S_{n,\alpha}$. Then there exists a constant $c$ such that $\min_{n,\hat{w}\in\mathcal{M}} \max_{S_{n,\alpha},\mathcal{D}\in\mathcal{D}_{\Sigma,\sigma^2,w^*,K},w^*,K} \mathbb{E}[\|\hat{w} - w^*\|_\Sigma^2] \geq c\,\alpha^2\,\sigma^2$.*

A proof is provided in App. I.1. A similar lower bound can be found in [11, Theorem 6.1].

## 4 Experimental results

### 4.1 DP Linear Regression

We present experimental results comparing our proposed technique (DP-ROBGD) with other baselines. We consider non-corrupted regression in this section and defer corrupted regression to the App. K. We begin by describing the problem setup and the baseline algorithms first.

**Experiment Setup.** We generate data for all the experiments using the following generative model. The parameter vector $w^*$ is uniformly sampled from the surface of a unit sphere. The covariates $\{x_i\}_{i=1}^n$ are first sampled from $\mathcal{N}(0, \Sigma)$ and then projected to unit sphere. We consider diagonal covariances $\Sigma$ of the following form: $\Sigma[0,0] = \kappa$, and $\Sigma[i,i] = 1$ for all $i \geq 1$. Here $\kappa \geq 1$ is the condition number of $\Sigma$. We generate noise $z_i$ from uniform distribution over $[-\sigma, \sigma]$. Finally, the response variables are generated as follows $y_i = x_i^\top w^* + z_i$. All the experiments presented below are repeated 5 times and the averaged results are presented. We set the DP parameters $(\epsilon, \delta)$ as $\epsilon = 1, \delta = \min(10^{-6}, n^{-2})$. Experiments for $\epsilon = 0.1$ can be found in Fig. 2 in the App. K.

**Baseline Algorithms.** We compare our estimator with the following baseline algorithms:

- *Non private algorithms:* ordinary least squares (OLS), one-pass stochastic gradient descent with tail-averaging (SGD). For SGD, step-size is $1/(2\lambda_{\max})$ and minibatch size is $n/T$, where $T = 3\kappa \log n$.

- *Private algorithms:* sufficient statistics perturbation (DP-SSP) [38, 83], differentially private stochastic gradient descent (DP-AMBSSGD) [81]. DP-SSP had the best empirical performance among numerous techniques studied by [83], and DP-AMBSSGD has the best known theoretical guarantees. The DP-SSP algorithm involves releasing $X^T X$ and $X^T \mathbf{y}$ differentially privately and computing $\widehat{(X^T X)}^{-1} \widehat{X^T \mathbf{y}}$. DP-AMBSSGD is a private version of SGD where the DP noise is set adaptively according to the excess error in each iteration. For both algorithms, we use the hyper-parameters recommended in their respective papers. To improve the performance of DP-AMBSSGD, we reduce the theoretical clipping threshold by a constant factor.

**DP-ROBGD.** We implement Alg. 1 with the following key changes. Instead of relying on PrivateNormEstimator to estimate $\Gamma$, we set it to its true value $\mathrm{Tr}(\Sigma)$. This is done for a fair comparison with DP-AMBSSGD which assumes the knowledge of $\mathrm{Tr}(\Sigma)$. Next, we use 20% of the samples to compute $\gamma_t$ in line 5 (instead of the 50% stated in Alg. 1). In our experiments we also present results for a variant of our algorithm called DP-ROBGD* which outputs the best iterate based on $\gamma_t$, instead of the last iterate. One could also perform tail-averaging instead of picking the best iterate. Both these modifications are primarily used to reduce the variance in the output of Alg. 1 and achieved similar performance in our experiments.

**Results.** Figure 1 presents the performance of various algorithms as we vary $n, \kappa, \sigma$. It can be seen that DP-ROBGD outperforms DP-AMBSSGD in almost all the settings (and DP-ROBGD* outperforms DP-ROBGD in all cases). DP-SSP has poor performance when the noise $\sigma$ is low, but performs slightly better than DP-ROBGD in other settings. A major drawback of DP-SSP is its computational complexity which scales as $O(nd^2 + d^\omega)$. In contrast, the computational complexity of DP-ROBGD has smaller dependence on $d$ and scales as $\tilde{O}(nd\kappa)$. Thus the latter is more computationally efficient for high-dimensional problems. More experimental results on both robust and private linear regression can be found in the App. K.

## 5 Sketch of the main ideas in the analysis

We provide the main ideas behind the proof of Thm. 3.1. The privacy proof is straightforward since no matter what clipping threshold we use the noise we add is always proportionally to the clipping threshold which guarantees privacy. In the remainder, we focus on the utility analysis.

The proof of the utility heavily relies on the *resilience* [73] (also known as *stability* [27]), which states that given a large enough sample set $S$, various statistics (for example, sample mean and sample variance) of any large enough subset of $S$ will be close to each other. We define resilience as follows.

**Definition 5.1** ([61, Definition 23]). For some $\alpha \in (0,1)$, $\rho_1 \in \mathbb{R}_+$, $\rho_2 \in \mathbb{R}_+$, and $\rho_3 \in \mathbb{R}_+$, $\rho_4 \in \mathbb{R}_+$, we say dataset $S_{\mathrm{good}} = \{(x_i \in \mathbb{R}^d, y_i \in \mathbb{R})\}_{i=1}^n$ is $(\alpha, \rho_1, \rho_2, \rho_3, \rho_4)$-resilient with respect to $(w^*, \Sigma, \sigma)$ for some $w^* \in \mathbb{R}^d$, positive definite $\Sigma \succ 0 \in \mathbb{R}^{d \times d}$, and $\sigma > 0$ if for any $T \subset S_{\mathrm{good}}$ of size $|T| \geq (1-\alpha)n$, the following holds for all $v \in \mathbb{R}^d$:

$$\left| \frac{1}{|T|} \sum_{(x_i, y_i) \in T} \langle v, x_i \rangle (y_i - x_i^\top w^*) \right| \leq \rho_1 \sqrt{v^\top \Sigma v}\, \sigma \ , \tag{5}$$

$$\left| \frac{1}{|T|} \sum_{x_i \in T} \langle v, x_i \rangle^2 - v^\top \Sigma v \right| \leq \rho_2 v^\top \Sigma v \ , \tag{6}$$

$$\left| \frac{1}{|T|} \sum_{(x_i, y_i) \in T} (y_i - x_i^\top w^*)^2 - \sigma^2 \right| \leq \rho_3 \sigma^2 \ , \tag{7}$$

$$\left| \frac{1}{|T|} \sum_{(x_i, y_i) \in T} \langle v, x_i \rangle \right| \leq \rho_4 \sqrt{v^\top \Sigma v} \ . \tag{8}$$

We give an overview of the proof for non-robust case as follows. First, we introduce some notations. Let $g_i^{(t)} := (x_i^\top w_t - y_i) x_i$ be the raw gradient and $\tilde{g}_i^{(t)} := \mathrm{clip}_\Theta(x_i) \mathrm{clip}_{\theta_t}(x_i^\top w_t - y_i)$ be the clipped gradient. Note that when the data follows from our distributional assumption, with high probability, samples are not clipped by the norm: $\mathrm{clip}_\Theta(x_i) = x_i$. We can write down one step of gradient update

(see Alg. 1) as follows:

$$w_{t+1} - w^* = \underbrace{\left(\mathbf{I} - \frac{\eta}{n}\sum_{i \in S} x_i x_i^\top\right)(w_t - w^*)}_{(i)} + \underbrace{\frac{\eta}{n}\sum_{i \in S} x_i z_i}_{(ii)} + \underbrace{\frac{\eta}{n}\sum_{i \in S}(g_i^{(t)} - \tilde{g}_i^{(t)})}_{(iii)} - \underbrace{\eta \phi_t \nu_t}_{(iv)} \ .$$

In the above equation, the first term is a contraction, meaning $w_t$ is moving toward $w^*$. The second term captures the noise from the randomness in the samples. The third term captures the bias introduced by the clipping operation, and the fourth term captures the added noise for privacy. The second term is standard and relatively easy to control, and our main focus is on the last two terms.

The third term $(\eta/n)\sum_{i \in S}(g_i^{(t)} - \tilde{g}_i^{(t)})$ can be controlled using the resilience property. We prove that with our estimated threshold, the clipping will only affect a small amount of datapoints, whose contribution to the gradient is small collectively.

Now we have controlled the deterministic bias. Then, we upper bound the fourth term, which is the noise for the purpose of privacy, and show the expected prediction error decrease in every gradient step. The difficulty is that, since our clipping threshold is adaptive, the decrease of the estimation error depends on the estimation error of all the previous steps. This causes that in some iterations, the estimation error actually increases. In order to get around this, we split the iterations into length $\kappa$ chunks, and argue that the maximum estimation error in a chunk must be a constant factor smaller than the previous chunk. This implies we will reach the desired error within $\tilde{O}(\kappa)$ steps.

## 6   Conclusion

We provide a novel variant of DP-SGD algorithm for differentially private linear regression under label corruption. We show the first near-optimal rate that achieves privacy and robustness to label corruptions simultaneously. When there is no label corruption, our result also improves upon the state-of-the-art method [81] in terms of the condition number $\kappa$. Compared to [81], our algorithm has two innovations: 1) we introduce a novel adaptive clipping, which is critical in achieving robustness against label corruptions; and 2) we use full batch gradient descent and a novel convergence analysis to get the near-optimal sample complexity.

## Acknowledgement

We thank Abhradeep Guha Thakurta for helpful discussions while working on this paper. This material is based upon work supported by the National Science Foundation under grants no. 2134012, 2019844, 2112471, and 2229876, and is supported in part by funds provided by the National Science Foundation, by the Department of Homeland Security, and by IBM. Any opinions, findings, and conclusions or recommendations expressed in this material are those of the author(s) and do not necessarily reflect the views of the National Science Foundation or its federal agency and industry partners.

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
