# Appendix

## A Related work

**Differentially private optimization.** There is a long line of work at the intersection of differentially privacy and optimization [17, 53, 13, 71, 12, 86, 7, 37, 72, 9, 58, 50, 87, 40, 39, 88]. As one of the most well-studied problem in differentially privacy, DP Empirical Risk Minimization (DP-ERM) aims to minimize the empirical risk $(1/n)\sum_{i \in S} \ell(x_i; w)$ privately. The optimal excess empirical risk for approximate DP (i.e., $\delta > 0$) is known to be $GD \cdot \sqrt{d}/(\varepsilon n)$, where the loss $\ell$ is convex and $G$-Lipschitz with respect to the data, and $D$ is the diameter of the convex parameter space [13]. This bound can be achieved by several DP-SGD methods, e.g., [71, 13], with different computational complexities. Differentially private stochastic convex optimization considers minimizing the population risk $\mathbb{E}_{x \sim \mathcal{D}}[\ell(x, w)]$, where data is drawn i.i.d. from some unknown distribution $\mathcal{D}$. Using some variations of DP-SGD, [12] and [37] achieves a population risk of $GD(1/\sqrt{n} + \sqrt{d}/(\varepsilon n))$.

**DP linear regression.** Applying above results for the linear model, by observing that $G = O(d)$ if $D = O(1)$, the sample complexity required for achieving generalization error is $n = d^2$. Existing works for DP linear regression, for example [82, 53, 65, 30, 85, 38, 64, 83, 70, 3] typically consider deterministic data. Under the i.i.d. Gaussian data setting, this translates into a sample complexity of $n = d^{3/2}/(\varepsilon\alpha)$, where the extra $d^{1/2}$ due to the fact that no statistical assumptions are made. For i.i.d. sub-Weibull data, recent work [81] achieved nearly optimal excess population risk $d/n + d^2/(\varepsilon^2 n^2)$ using DP-SGD with adaptive clipping, up to extra factors on the condition number. This is closest to our work and we provide detailed comparisons in Sections 2.1 and 3.2. Under Gaussian assumptions, [63] analyze linear regression algorithm with sub-optimal guarantees. [32, 6, 5, 61] also consider using robust statistics like Tukey median [79] or Theil–Sen estimator [77] for differentially private regression. However, [32] and [6] lack utility guarantees and [5] is restricted to one-dimensional data. [61] achieves optimal sample complexity but takes exponential time. More recently, [20] uses private linear regression scenario to show that correlated noise provably improves upon vanilla DP-SGD.

Recent work [16] considers DP generalized linear model and provides a DP-SGD type algorithm that achieves nearly optimal error $d/n + d^2/(\varepsilon^2 n^2)$. Their result is not comparable to ours because they assume the norm of the gradient is bounded by a constant, while for linear regression, the norm of the gradient is $O(\sqrt{d})$.

**Robust linear regression.** Robust mean estimation and linear regression have been studied for a long time in the statistics community [80, 46, 79]. However, for high dimensional data, these estimators generalizing the notion of median to higher dimensions are typically computationally intractable. Recent advances in the filter-based algorithms, e.g., [26, 23, 24, 28, 18, 31], achieve nearly optimal guarantees for mean estimation in time linear in the dimension of the dataset. Motivated by the filter algorithms, [29, 25, 68, 67, 19, 47] achieved nearly optimal rate with $d$ samples for robust linear regression, where both data $x_i$ and label $y_i$ are corrupted. Another type of efficient methods that achieve similar rates and sample complexity in polynomial time is based on sum-of-square proofs [54, 11], which can be computationally expensive in practice. [89] and [61] achieves nearly optimal rates using $d$ samples but require exponential time complexities. An important special case of adversarial corruption is when the adversary only corrupts the response variable in supervised learning [52] and also in unsupervised learning [78]. For linear regression, when there is only label corruptions, [15, 21, 55] achieve nearly optimal rates with $O(d)$ samples. Under the oblivious label corruption model, i.e., the adversary only corrupts a fraction of labels in complete ignorance of the data, [14, 75] provide consistent estimator $\hat{w}_n$ such that $\lim_{n \to \infty} \mathbb{E}[\hat{w}_n - w^*]_2 = 0$ with $O(d)$ samples.

Of these, [15, 21] are relevant to our work as they consider the same adversary model as Asmp. 3.7. When $x_i$'s and $z_i$'s are sampled from $\mathcal{N}(0, \Sigma)$ and $\mathcal{N}(0, \sigma^2)$, [21] proposed a Huber loss based estimator that achieves error rate of $\sigma^2\alpha^2 \log^2(n/\delta)$ when $n = \tilde{O}\left(\kappa^2 d/\alpha^2\right)$. Under the same setting, [15] proposed a hard thresholding based estimator that achieves $\sigma^2\alpha^2$ error rate with $\tilde{O}\left(d/\alpha^2\right)$ sample complexity. Our results in Thm. 3.1 match these rates, except for the sub-optimal dependence on $\log^2(1/\alpha)$. Another line of work considered both label and covariate corruptions and developed optimal algorithms for parameter recovery [29, 25, 68, 67, 19, 47, 54, 11, 89, 22]. The best existing

efficient algorithm , e.g. [67], achieves error rate of $\sigma^2 \alpha^2 \log(1/\alpha)$ when $n = \tilde{O}\left(d/\alpha^2\right)$, and the $x_i$ and $z_i$ are sampled from $\mathcal{N}(0, I)$ and $\mathcal{N}(0, \sigma^2)$.

**Robust and private linear regression.** Under the settings of both DP and data corruptions, the only algorithm by [61] achieves nearly optimal rates $\alpha \log(1/\alpha)\sigma$ with optimal sample complexities of $d/\alpha^2 + d/(\varepsilon\alpha)$. However, their algorithm requires exponential time complexities.

**Robust and private mean estimation** Based on sum-of-square proofs, recent works [45, 4] are able to achieve nearly optimal rates $\alpha \log(1/\alpha)$ with $\tilde{O}(d)$ samples for sub-Gaussian data with known covariance.

# B Preliminary on differential privacy

Our algorithm builds upon two DP primitive: Gaussian mechanism and private histogram. The Gaussian mechanism is one examples of a larger family of mechanisms known as output perturbation mechanisms. In practice, it is possible to get better utility trade-off for a output perturbation mechanism by carefully designing the noise, such as the stair-case mechanism which are shown to achieve optimal utility in the variance [41] and also in hypothesis testing [48]. However, the gain is only by constant factors, which we do not try to optimize in this paper. We provide a reference for the Gaussian mechanism and private histogram below.

**Lemma B.1** (Gaussian mechanism [34]). *For a query $q$ with sensitivity $\Delta_q$, the Gaussian mechanism outputs $q(S) + \mathcal{N}(0, (\Delta_q\sqrt{2\log(1.25/\delta)}/\varepsilon)^2 \mathbf{I}_d)$ and achieves $(\varepsilon, \delta)$-DP.*

**Lemma B.2** (Stability-based histogram [51, Lemma 2.3]). *For every $K \in \mathbb{N} \cup \{\infty\}$, domain $\Omega$, for every collection of disjoint bins $B_1, \ldots, B_K$ defined on $\Omega$, $n \in \mathbb{N}$, $\varepsilon \geq 0, \delta \in (0, 1/n)$, $\beta > 0$ and $\alpha \in (0, 1)$ there exists an $(\varepsilon, \delta)$-differentially private algorithm $M : \Omega^n \to \mathbb{R}^K$ such that for any set of data $X_1, \ldots, X_n \in \Omega^n$*

1. *$\hat{p}_k = \frac{1}{n}\sum_{X_i \in B_k} 1$*

2. *$(\tilde{p}_1, \ldots, \tilde{p}_K) \leftarrow M(X_1, \ldots, X_n)$, and*

3.
$$n \geq \min\left\{\frac{8}{\varepsilon\beta}\log(2K/\alpha), \frac{8}{\varepsilon\beta}\log(4/\alpha\delta)\right\}$$

*then,*
$$\mathbb{P}(|\tilde{p}_k - \hat{p}_k| \leq \beta) \geq 1 - \alpha$$

When the databse is accessed multiple times, we use the following composition theorems to account for the end-to-end privacy leakage.

**Lemma B.3** (Parallel composition [62]). *Consider a sequence of interactive queries $\{q_k\}_{k=1}^K$ each operating on a subset $S_k$ of the database and each satisfying $(\varepsilon, \delta)$-DP. If $S_k$'s are disjoint then the composition $(q_1(S_1), q_2(S_2), \ldots, q_K(S_K))$ is $(\varepsilon, \delta)$-DP.*

**Lemma B.4** (Serial composition [34]). *If a database is accessed with an $(\varepsilon_1, \delta_1)$-DP mechanism and then with an $(\varepsilon_2, \delta_2)$-DP mechanism, then the end-to-end privacy guarantee is $(\varepsilon_1 + \varepsilon_2, \delta_1 + \delta_2)$-DP.*

In most modern privacy analysis of iterative processes, advanced composition theorem from [49] gives tight accountant for the end-to-end privacy budget. It can be improved for specific mechanisms using tighter accountants, e.g., in [66, 42, 84, 90, 43].

**Lemma B.5** (Advanced composition [49]). *For $\varepsilon \leq 0.9$, an end-to-end guarantee of $(\varepsilon, \delta)$-differential privacy is satisfied if a database is accessed $k$ times, each with a $(\varepsilon/(2\sqrt{2k\log(2/\delta)}), \delta/(2k))$-differential private mechanism.*

# C Adaptive clipping for the gradient norm

In the ideal clipping thresholds for the norm and the residual, there are unknown terms which we need to estimate adaptively, $(\|w_t - w^*\|_{\Sigma}^2 + \sigma^2)$ and $\mathrm{Tr}(\Sigma)$, up to constant multiplicative errors. We privately estimate the (squared and shifted) distance to optimum, $(\|w_t - w^*\|_{\Sigma}^2 + \sigma^2)$, with Alg. 2

and privately estimate the average input norm, $\mathbb{E}[\|x_i\|^2] = \text{Tr}(\Sigma)$, with Alg. 3 in App. F. These are used to get the clipping thresholds in Alg. 1. We propose a trimmed mean approach below for distance estimation. The norm estimator is similar and is provided in App. F.

**Private distance estimation using private trimmed mean.** The goal is to estimate the (shifted) distance to optimum, $\|w_t - w^*\|_\Sigma^2 + \sigma^2$, up to some constant multiplicative error. Note that this is precisely the task of estimating the variance of the residual $b_i = y_i - w_t^\top x_i$. When there is no adversarial corruption and no privacy constraint, we can simply use the empirical variance estimator $(1/n) \sum_{i \in [n]} (y_i - w_t^\top x_i)^2$ to obtain a good estimate. However, the empirical variance estimator is not robust against adversarial corruptions since one outlier can make the estimate arbitrarily large. A classical idea is using the *trimmed estimator* from [80], which throws away the $2\alpha$ fraction of residuals $b_i$ with the largest magnitude. For datasets with resilience property as assumed in this paper, this will guarantee an accurate estimate of the distance to optimum in the presence of $\alpha$ fraction of corruptions.

To make the estimator private, it is tempting to simply add a Laplacian noise to the estimate. However, the sensitivity of the trimmed estimator is unknown and depends on the distance to the optimum that we aim to estimate; this makes it challenging to determine the variance of the Laplacian noise we add. Instead, we propose to partition the dataset into $k$ batches, compute an estimate for each batch, and form a histogram with over those $k$ estimates. Using a private histogram mechanism with geometrically increasing bin sizes, we propose using the bin with the most estimates to guarantee a constant factor approximation of the distance to the optimum. We describe the algorithm as follows.

---

**Algorithm 2:** Robust and Private Distance Estimator

**Input:** $S_2 = \{(x_i, y_i)\}_{i=1}^n$, current $w_t$, $(\varepsilon_0, \delta_0)$, failure probability $\zeta$,

1 Let $b_i \leftarrow (y_i - w_t^\top x_i)^2, \forall i \in [n]$ and $\tilde{S} \leftarrow \{b_i\}_{i=1}^n$.

2 Partition $\tilde{S}$ into $k = \lceil C_1 \log(1/(\delta_0 \zeta))/\varepsilon_0 \rceil$ subsets of equal size and let $G_j$ be the $j$-th partition.

3 For $j \in [k]$, denote $\psi_j$ as the 0.9-quantile of $G_j$ and $\phi_j \leftarrow \frac{1}{|G_j|} \sum_{i \in G_j} b_i \mathbf{1}\{b_i \leq \psi_j\}$.

4 Partition $[0, \infty)$ into geometrically increasing intervals
$\Omega := \left\{ \ldots, \left[ 2^{-1}, 1 \right), [1, 2), \left[ 2, 2^2 \right), \ldots \right\} \cup \{[0, 0]\}$

5 Run $(\varepsilon_0, \delta_0)$-DP histogram of Lemma B.2 on $\{\phi_j\}_{j=1}^k$ over $\Omega$

6 **if** all the bins are empty **then** Return $\perp$

7 Let $[\ell, r]$ be a non-empty bin that contains the maximum number of points in the DP histogram

8 **return** $\ell$

---

This algorithm gives an estimate of the distance up to a constant multiplicative error as we show in the following theorem. We provide a proof in App. D.

**Theorem C.1.** *Alg. 2 is $(\varepsilon_0, \delta_0)$-DP. For an $\alpha_{\text{corrupt}}$-corrupted dataset $S_2$ that satisfy Asmp. 2.1 and Asmp. 3.7 and any $\zeta \in (0, 1)$, if*

$$n = O\left( \frac{(d + \log((\log(1/(\delta_0 \zeta)))/\varepsilon_0 \zeta))(\log(1/(\delta_0 \zeta)))}{\varepsilon_0} \right), \tag{9}$$

*with a large enough constant, then with probability $1 - \zeta$, Alg. 2 returns $\ell$ such that $\frac{1}{4}(\|w_t - w^*\|_\Sigma^2 + \sigma^2) \leq \ell \leq 4(\|w_t - w^*\|_\Sigma^2 + \sigma^2)$.*

Note that in Thm. C.1, we only need to estimate distance up to a constant multiplicative error, as opposed to an error that depends on our final end-to-end desired level $\alpha$. Consequently, we require smaller sample complexity (that doesn't depend on $\alpha$) than other parts of our approach.

*Remark* C.2. While DP-STAT (Algorithm 3 in [81]) can also be used to estimate $\|w_t - w^*\|_\Sigma + \sigma$ (and it would not change the ultimate sample complexity in its dependence on $\kappa$, $d$, $\varepsilon$, and $n$), there are three important improvements we make: $(i)$ DP-STAT requires the knowledge of $\|w^*\|_\Sigma + \sigma$; $(ii)$ our utility guarantee has improved dependence in $K$ and $\log(n)$; and $(iii)$ Alg. 2 is robust against label corruption.

**Upper bound on clipped good data points.** Using the above estimated distance to the optimum in selecting a threshold $\theta_t$, we also need to ensure that we do not clip too many clean data points. The

tolerance in our algorithm to reach the desired level of accuracy is clipping $O(\alpha)$ fraction of clean data points. This is ensured by the following lemma, and we provide a proof in App. E.

**Lemma C.3.** *Under Asmp. 2.1 and for all $t \in [T]$, if $\theta_t \geq \sqrt{9C_2K^2 \log(1/(2\alpha))} \cdot (\|w^* - w_t\|_\Sigma + \sigma)$ then $\left|\{i \in S_3 \cap S_{\text{good}} : |w_t^\top x_i - y_i| \geq \theta_t\}\right| \leq \alpha n$.*

## D  Proof of Thm. C.1 on the private distance estimation

We present our formal theorem for the general sub-Weibull distribution as follows.

**Theorem D.1.** *Alg. 2 is $(\varepsilon_0, \delta_0)$-DP. For an $\alpha_{\text{corrupt}}$-corrupted dataset $S_2$ satisfying Asmp. H.1 and Asmp. 3.7 and an upper bound $\bar{\alpha}$ on $\alpha_{\text{corrupt}}$ that satisfy $37C_2K^2 \cdot \bar{\alpha} \log^{2a}(1/(6\bar{\alpha})) \leq 1/4$ and any $\zeta \in (0, 1)$, if*

$$n = O\left(\frac{(d + \log((\log(1/(\delta_0\zeta)))/\varepsilon_0\zeta))(\log(1/(\delta_0\zeta)))}{\bar{\alpha}^2\varepsilon_0}\right), \tag{10}$$

*with a large enough constant then, with probability $1 - \zeta$, Alg. 2 returns $\ell$ such that $\frac{1}{4}(\|w_t - w^*\|_\Sigma^2 + \sigma^2) \leq \ell \leq 4(\|w_t - w^*\|_\Sigma^2 + \sigma^2)$.*

We first analyze the privacy. Changing a data point $(x_i, y_i)$ can affect at most one partition in $\{G_j\}_{j=1}^k$. This would affect at most two histogram bins, increasing the count of one bin by one and decreasing the count in another bin by one. Under such a bounded $\ell_1$ sensitivity, the privacy guarantees follows from Lemma B.2.

Next, we analyze the utility. In the (private) histogram step, we claim that at most only two consecutive bins can be occupied by any $\phi_j$'s. This is also true for the private histogram, because the private histogram of Lemma B.2 adds noise to non-empty bins only. By Lemma B.2, if $k \geq c \log(1/(\delta_0\zeta_0))/\varepsilon_0$, one of these two intervals (the union of which contains the true distance $\|w_t - w^*\|_\Sigma^2 + \sigma^2$) is released. This results in a multiplicative error bound of four, as the bin size increments by a factor of two.

To show that only two bins are occupied, we show that all $\phi_j$'s are close to the true distance. We first show that each partition contains at most $2\alpha_{\text{corrupt}}$ fraction of corrupted samples and thus all partitions are $(2\bar{\alpha}, 6\bar{\alpha}, 6\hat{\rho}, 6\hat{\rho}, 6\hat{\rho}, 6\hat{\rho}')$-corrupt good, where $\hat{\rho}(C_2, K, a, \bar{\alpha}) = C_2K^2\bar{\alpha} \log^{2a}(1/6\bar{\alpha})$ and $\hat{\rho}'(C_2, K, a, \bar{\alpha}) = C_2K\bar{\alpha} \log^a(1/6\bar{\alpha})$, as defined in Definition J.6.

Let $B = \lfloor n/k \rfloor$ be the sample size in each partition. Let $\zeta_0 = \zeta/2$. Since the partition is drawn uniformly at random, for each partition $G_j$, the number of corrupted samples $\alpha'n$ satisfies $\alpha'n \sim$ Hypergeometric$(n, \alpha_{\text{corrupt}}n, n/k)$. The tail bound gives that with probability $1 - \zeta_0$,

$$\alpha' \leq \alpha_{\text{corrupt}} + (k/n)\log(2/\zeta_0) \leq 2\bar{\alpha},$$

where the last inequality follows from the fact that the corruption level is bounded by $\alpha_{\text{corruption}} \leq \bar{\alpha}$ and the assumption on the sample size in Eq. (10) which implies $n \gtrsim \log(1/(\delta_0\zeta_0))\log(1/\zeta_0)/(\bar{\alpha}\varepsilon_0)$.

For a particular subset $G_j$, Lemma J.7 implies that if $B = O((d + \log(1/\zeta_0))/\bar{\alpha}^2)$, then $G_j$ is $(\alpha', 6\bar{\alpha}, 6\hat{\rho}, 6\hat{\rho}, 6\hat{\rho}')$-corrupt good set with respect to $(w^*, \Sigma, \sigma)$ from Asmp. H.1. This means that there exists a constant $C_2 > 0$ such that for any $T_1 \subset S_{\text{good}}$ with $|T_1| \geq (1 - 6\bar{\alpha})B$, we have

$$\left|\frac{1}{|T_1|}\sum_{i \in T_1}\langle x_i, w^* - w_t\rangle^2 - \|w^* - w_t\|_\Sigma^2\right| \leq 6C_2K^2\bar{\alpha}\log^{2a}(1/(6\bar{\alpha}))\|w^* - w_t\|_\Sigma^2,$$

$$\left|\frac{1}{|T_1|}\sum_{i \in T_1}z_i^2 - \sigma^2\right| \leq 6C_2K^2\bar{\alpha}\log^{2a}(1/(6\bar{\alpha}))\sigma^2,$$

and

$$\left|\frac{1}{|T_1|}\sum_{i \in T_1}z_i\langle x_i, w^* - w_t\rangle\right| \leq 6C_2K^2\bar{\alpha}\log^{2a}(1/(6\bar{\alpha}))\|w^* - w_t\|_\Sigma\sigma.$$

Note that for $i \in S_{\text{good}}$, $b_i = z_i^2 + 2z_i(w^* - w_t)^\top x_i + (w^* - w_t)^\top x_i x_i^\top (w^* - w_t)$. By the triangular inequality, we know, under above conditions,

$$\left| \frac{1}{|T_1|} \sum_{i \in T_1} b_i - \|w^* - w_t\|_\Sigma^2 - \sigma^2 \right| \leq 12 C_2 K^2 \bar{\alpha} \log^{2a}(1/(6\bar{\alpha}))(\|w^* - w_t\|_\Sigma^2 + \sigma^2) . \quad (11)$$

Which also implies that any subset $T_2 \subset S_{\text{good}}$ and $|T_2| \leq 6\bar{\alpha}|S_{\text{good}}|$, we have

$$\left| \frac{1}{|T_2|} \sum_{i \in T_2} b_i - \|w^* - w_t\|_\Sigma^2 - \sigma^2 \right| \leq 12 C_2 K^2 \log^{2a}(1/(6\bar{\alpha}))(\|w^* - w_t\|_\Sigma^2 + \sigma^2) . \quad (12)$$

Recall that $\psi_j$ is the $(1 - 3\bar{\alpha})$-quantile of the dataset $G_j$. Let $T := \{i \in S_{\text{good}} : b_i \leq \psi_j\}$, where with a slight abuse of notations, we use $S_{\text{good}}$ to denote the set of uncorrupted samples corresponding to $G_j$ and $S_{\text{bad}}$ to denote the set of corrupted samples corresponding to $G_j$. Since the corruption is less than $\alpha'$, we know $(1 - 3\bar{\alpha} - \alpha')B \leq |T| \leq (1 - 3\bar{\alpha} + \alpha')B$. By our assumption that $\alpha' \leq 2\bar{\alpha}$, we have $|\bar{E}| \geq (3\bar{\alpha} - \alpha')B \geq \bar{\alpha}B$ where $\bar{E} := S_{\text{good}} \setminus E$. Using Eq. (12) with a choice of $T_2 = \bar{E}$, we get that

$$\min_{i \in \bar{E}} b_i - \|w^* - w_t\|_\Sigma^2 - \sigma^2 \leq 12 C_2 K^2 \log^{2a}(1/(6\bar{\alpha}))(\|w^* - w_t\|_\Sigma^2 + \sigma^2) . \quad (13)$$

This implies that

$$\psi_j \leq 12 C_2 K^2 \log^{2a}(1/(6\bar{\alpha}))(\|w^* - w_t\|_\Sigma^2 + \sigma^2). \quad (14)$$

Hence

$$\left| \phi_j - \|w^* - w_t\|_\Sigma^2 - \sigma^2 \right| = \left| \frac{1}{B} \sum_{i \in G_j} b_i \cdot \mathbf{1}\{b_i \leq \psi_j\} - \|w^* - w_t\|_\Sigma^2 - \sigma^2 \right|$$

$$= \left| \frac{1}{B} \sum_{i \in T} b_i - \|w^* - w_t\|_\Sigma^2 - \sigma^2 \right| + \left| \frac{1}{B} \sum_{i \in S_{\text{bad}}} b_i \cdot \mathbf{1}\{b_i \leq \psi_j\} \right|$$

$$\leq 37 C_2 K^2 \cdot \bar{\alpha} \log^{2a}(1/(6\bar{\alpha}))(\|w^* - w_t\|_\Sigma^2 + \sigma^2), \quad (15)$$

where we applied Eq. (14) and Eq. (11) in the last inequality.

On a fixed partition $G_j$, we showed that if $B = O((d + \log(1/\zeta_0))/\bar{\alpha}^2)$ then, with probability $1 - \zeta_0$, $|\phi_j - \|w^* - w_t\|_\Sigma^2 - \sigma^2| \leq \frac{1}{4}(\|w^* - w_t\|_\Sigma^2 + \sigma^2)$, which follows from our assumption that $37 C_2 K^2 \cdot \bar{\alpha} \log^{2a}(1/(6\bar{\alpha})) \leq 1/4$. Using an union bound for all subsets, we know if $B = O((d + \log(k/\zeta_0))/\bar{\alpha}^2)$, then $1 - \zeta_0$, $|\phi_j - \|w^* - w_t\|_\Sigma^2 - \sigma^2| \leq \frac{1}{4}(\|w^* - w_t\|_\Sigma^2 + \sigma^2)$ holds for all $j \in [k]$. Since the upper bound lower bound ratio is $5/3$ which is less than $2$. All the $\phi_j$ must lie in two bins, which will result in a factor of $4$ multiplicative error.

## E  Proof of Lemma C.3 on the upper bound on clipped good points

Let $\hat{\rho}(C_2, K, a, \alpha) = 2 C_2 K^2 \alpha \log^{2a}(1/(2\alpha))$ and $\hat{\rho}'(C_2, K, a, \alpha) = 2 C_2 K \alpha \log^a(1/(2\alpha))$. Lemma J.7 implies that if $n = O((d + \log(1/\zeta))/(\alpha^2))$ with a large enough constant, then there exists a universal constant $C_2$ such that $S_3$ is, with respect to $(w^*, \Sigma, \sigma)$, $(\alpha_{\text{corrupt}}, 2\alpha, \hat{\rho}, \hat{\rho}, \hat{\rho}, \hat{\rho}')$-corrupt good. The rest of the proof is under this (deterministic) resilience condition. By the resilience property in Eq. (6), we know for any $T \subset S_{\text{good}}$ with $|T| \geq (1 - 2\alpha)n$,

$$\left| \frac{1}{|T|} \sum_{i \in T} (w^* - w_t)^\top x_i x_i^\top (w^* - w_t) - \|w^* - w_t\|_\Sigma^2 \right| \leq 2 C_2 K^2 \alpha \log^{2a}(1/(2\alpha)) \|w^* - w_t\|_\Sigma^2 . \tag{16}$$

Let $E := \{i \in S_{\text{good}} : (w^* - w_t)^\top x_i x_i^\top (w^* - w_t) > \|w^* - w_t\|_\Sigma^2 (8 C_2 K^2 \log^{2a}(1/(2\alpha)) + 1)\}$. Denote $\tilde{\alpha} := |E|/n$. We want to show that $\tilde{\alpha} \leq \alpha/2$. Let $T$ be the set of points that contain the smallest $1 - \alpha/2$ fraction in $\{(w^* - w_t)^\top x_i x_i^\top (w^* - w_t)\}_{i \in S_{\text{good}}}$. We know $|T| = (1 - \alpha/2)n \geq$

$(1-2\alpha)n$. To prove by contradiction, suppose $\tilde{\alpha} > \alpha/2$, which means all data points in $S_{\text{good}} \setminus T$ are larger than $\|w^* - w_t\|_\Sigma^2 (8C_2K^2 \log^{2a}(1/(2\alpha)) + 1)$. From resilience property in Eq. (16), we know

$$\frac{1}{n} \sum_{i \in S_{\text{good}}} (w^* - w_t)^\top x_i x_i^\top (w^* - w_t)$$

$$= \frac{1}{n} \sum_{i \in T} (w^* - w_t)^\top x_i x_i^\top (w^* - w_t) + \frac{1}{n} \sum_{i \in S_{\text{good}} \setminus T} (w^* - w_t)^\top x_i x_i^\top (w^* - w_t)$$

$$\geq \left(1 - \frac{\alpha}{2}\right) \left(1 - 2C_2K^2\alpha \log^{2a}\left(\frac{1}{2\alpha}\right)\right) \|w^* - w_t\|_\Sigma^2 + \frac{\alpha}{2}(8C_2K^2 \log^{2a}(\frac{1}{2\alpha}) + 1)\|w^* - w_t\|_\Sigma^2$$

$$> (1 + 2C_2K^2\alpha \log^{2a}(1/2\alpha))\|w^* - w_t\|_\Sigma^2 \, ,$$

which contradicts Eq. (16) for $S_{\text{good}}$. This shows $\tilde{\alpha} \leq \alpha/2$.

Similarly, we can show that $\left|\left\{i \in S_{\text{good}} : z_i^2 > \sigma^2(8C_2K^2 \log^{2a}(1/(2\alpha)) + 1)\right\}\right| \leq \alpha/2$. This means the rest $(1-\alpha)n$ points in $S_{\text{good}}$ satisfies $\sqrt{(w^* - w_t)^\top x_i x_i^\top (w^* - w_t)} + |z_i| \leq (\|w_t - w^*\| + \sigma)\sqrt{(8C_2K^2 \log^{2a}(1/(2\alpha)) + 1)}$. Note that for all $i \in S_{\text{good}}$, we have

$$|x_i^\top w_t - y_i| = |x_i^\top (w_t - w^*) - z_i|$$
$$\leq |x_i^\top (w_t - w^*)| + |z_i|$$
$$\leq \left(\sqrt{(w^* - w_t)^\top x_i x_i^\top (w^* - w_t)} + |z_i|\right) \, .$$

By our assumption that $C_2K^2 \log^{2a}(1/(2\bar{\alpha})) \geq 1$ which follows from Asmp. 3.7, we have

$$\left|\left\{i \in S_{\text{good}} : |x_i^\top w_t - y_i| \leq (\|w_t - w^*\|_\Sigma + \sigma)\sqrt{9C_2K^2 \log^{2a}(1/(2\alpha))}\right\}\right| \geq (1-\alpha)n \, . \quad (17)$$

## F   Private norm estimation: algorithm and analysis

---

**Algorithm 3:** Private Norm Estimator

---

**Input:** $S_1 = \{(x_i, y_i)\}_{i=1}^n$, target privacy $(\varepsilon_0, \delta_0)$, failure probability $\zeta$.

1 Let $a_i \leftarrow \|x_i\|^2$. Let $\tilde{S} = \{a_i\}_{i=1}^n$.

2 Partition $\tilde{S}$ into $k = \lfloor C_1 \log(1/(\delta_0\zeta))/\varepsilon \rfloor$ subsets of equal size and let $G_j$ be the $j$-th partition.

3 For each $j \in [k]$, denote $\psi_j = (1/|G_j|) \sum_{i \in G_j} a_i$.

4 Partition $[0, \infty)$ into bins of geometrically increasing intervals
   $\Omega := \left\{\ldots, \left[2^{-2/4}, 2^{-1/4}\right), \left[2^{-1/4}, 1\right), \left[1, 2^{1/4}\right), \left[2^{1/4}, 2^{2/4}\right), \ldots\right\} \cup \{[0, 0]\}$

5 Run $(\varepsilon_0, \delta_0)$-DP histogram learner of Lemma B.2 on $\{\psi_j\}_{j=1}^k$ over $\Omega$

6 **if** all the bins are empty **then** Return $\perp$

7 Let $[\ell, r]$ be a non-empty bin that contains the maximum number of points in the DP histogram

8 Return $\ell$

---

**Lemma F.1.** *Alg. 3 is $(\varepsilon_0, \delta_0)$-DP. If $\{x_i\}_{i=1}^n$ are i.i.d. samples from $(K, a)$-sub-Weibull distributions with zero mean and covariance $\Sigma$ and*

$$n = \tilde{O}\left(\frac{\log^{2a}(1/(\delta_0\zeta))}{\varepsilon_0}\right) \, ,$$

*with a large enough constant then Alg. 3 returns $\Gamma$ such that, with probability $1 - \zeta$,*

$$\frac{1}{\sqrt{2}} \operatorname{Tr}(\Sigma) \leq \Gamma \leq \sqrt{2} \operatorname{Tr}(\Sigma) \, .$$

We provide a proof in App. F.1.

### F.1 Proof of Lemma F.1 on the private norm estimation

By Hanson-Wright inequality in Lemma J.1 and union bound, there exists constant $c > 0$ such that with probability $1 - \zeta$,

$$|\frac{1}{b} \sum_{i=1}^{b} \|x_i\|^2 - \text{Tr}(\Sigma)| \leq cK^2 \text{Tr}(\Sigma) \left( \sqrt{\frac{\log(1/\zeta)}{b}} + \frac{\log^{2a}(1/\zeta)}{b} \right) , \tag{18}$$

This means there exists a constant $c' > 0$ such that if $b \geq c'K^2 \log^{2a}(k/\zeta)$, then for all $j \in [k]$.

$$|\psi_j - \text{Tr}(\Sigma)| \leq 2^{1/8} \text{Tr}(\Sigma) \tag{19}$$

With probability $1 - \zeta$, $\{\psi_j\}_{j=1}^{k}$ lie in interval of size $2^{1/4} \text{Tr}(\Sigma)$. Thus, at most two consecutive bins are filled with $\{\psi_j\}_{j=1}^{k}$. Denote them as $I = I_1 \cup I_2$. Our analysis indicates that $\mathbb{P}(\psi_i \in I) \geq 0.99$. By private histogram in Lemma B.2, if $k \geq \log(1/(\delta\zeta))/\varepsilon$, $|\hat{p}_I - \tilde{p}_I| \leq 0.01$ where $\hat{p}_I$ is the empirical count on $I$ and $\tilde{p}_I$ is the noisy count on $I$. Under this condition, one of these two intervals are released. This results in multiplicative error of $\sqrt{2}$.

## G   Proof of the resilience in Lemma J.7

We apply following resilience property for general distribution characterized by Orlicz function from [89].

**Lemma G.1** ([89, Theorem 3.4]). *Dataset $S = \{x_i \in \mathbb{R}^d\}_{i=1}^{n}$ consists i.i.d. samples from a distribution $\mathcal{D}$. Suppose $\mathcal{D}$ is zero mean and satisfies $\mathbb{E}_{x \sim \mathcal{D}} \left[ \psi \left( \frac{(v^\top x)^2}{\kappa^2 \mathbb{E}_{x \sim \mathcal{D}}[(v^\top x)^2]} \right) \right] \leq 1$ for all $v \in \mathbb{R}^d$, where $\psi(\cdot)$ is Orlicz function. Let $\Sigma = \mathbb{E}_{x \sim \mathcal{D}}[xx^\top]$. Suppose $\alpha \leq \bar{\alpha}$, where $\bar{\alpha}$ satisfies $(1 + \bar{\alpha}/2) \cdot 2\kappa^2 \bar{\alpha} \psi^{-1}(2/\bar{\alpha}) < 1/3$, $\bar{\alpha} \leq 1/4$. Then there exists constant $c_1, C_2$ such that if $n \geq c_1((d + \log(1/\zeta))/(\alpha^2))$, with probability $1 - \zeta$, for any $T \subset S$ of size $|T| \geq (1 - \alpha)n$, the following holds:*

$$\left\| \Sigma^{-1/2} \left( \frac{1}{|T|} \sum_{i \in T} x_i \right) \right\| \leq C_2 \kappa \alpha \sqrt{\psi^{-1}(1/\alpha)} \tag{20}$$

*and*

$$\left\| \mathbf{I}_d - \Sigma^{-1/2} \left( \frac{1}{|T|} \sum_{i \in T} x_i x_i^\top \right) \Sigma^{-1/2} \right\|_2 \leq C_2 \kappa^2 \alpha \psi^{-1}(1/\alpha) . \tag{21}$$

Let $\psi(t) = e^{t^{1/(2a)}}$. It is easy to see that $\psi(t)$ is a valid Orlicz function. Then if $x_i$ is $(K, a)$-sub-Weibull, then we know

$$\left\| \Sigma^{-1/2} \left( \frac{1}{|T|} \sum_{i \in T} x_i \right) \right\| \leq C_2 K \alpha \sqrt{\log^{2a}(1/\alpha)} , \tag{22}$$

and

$$\left\| \mathbf{I}_d - \Sigma^{-1/2} \left( \frac{1}{|T|} \sum_{i \in T} x_i x_i^\top \right) \Sigma^{-1/2} \right\|_2 \leq C_2 K^2 \alpha \log^{2a}(1/\alpha) . \tag{23}$$

This implies

$$(1 - C_2 K^2 \alpha \log^{2a}(1/\alpha))\mathbf{I}_d \preceq \Sigma^{-1/2} \left( \frac{1}{|T|} \sum_{i \in T} x_i x_i^\top \right) \Sigma^{-1/2} \preceq (1 + C_2 K^2 \alpha \log^{2a}(1/\alpha))\mathbf{I}_d . \tag{24}$$

Using the fact that $C^\top A C \preceq C^\top B C$ if $A \preceq B$, we know

$$(1 - C_2 K^2 \alpha \log^{2a}(1/\alpha))\Sigma \preceq \frac{1}{|T|} \sum_{i \in T} x_i x_i^\top \preceq (1 + C_2 K^2 \alpha \log^{2a}(1/\alpha))\Sigma . \tag{25}$$

This implies resilience properties of $x_i$ and $z_i$ in Eq. (6) and Eq. (7) in Definition 5.1 respectively. Next, we show the resilience property of $x_i z_i$.

By $ab \leq \frac{a^2}{2} + \frac{b^2}{2}$, for any fixed $v \in \mathbb{R}^d$,

$$\mathbb{E}[\exp\left(\left(\frac{|\langle x_i z_i, v\rangle|^2}{K^4 \sigma^2 v^\top \Sigma v}\right)^{1/(4a)}\right)] \leq \mathbb{E}\left[\exp\left(\left(\frac{|\langle x_i, v\rangle|^2}{K^2 v^\top \Sigma v}\right)^{1/(2a)}/2\right)\exp\left(\left(\frac{z_i^2}{K^2 \sigma^2}\right)^{1/(2a)}/2\right)\right] \tag{26}$$

$$\leq \frac{1}{2}\left(\mathbb{E}\left[\exp\left(\left(\frac{|\langle x_i, v\rangle|^2}{K^2 v^\top \Sigma v}\right)^{1/(2a)}\right)\right] + \mathbb{E}\left[\exp\left(\left(\frac{z_i^2}{K^2 \sigma^2}\right)^{1/(2a)}\right)\right]\right) \tag{27}$$

$$\leq 2 . \tag{28}$$

Since $\mathbb{E}[x_i z_i] = 0$, [89, Lemma E.3] implies that there exists constant $c_1, C_2 > 0$ such that if $n \geq c_1(d + \log(1/\zeta))/(\alpha^2)$, with probability $1 - \zeta$, for any $T \subset S_{\text{good}}$ of size $|T| \geq (1 - \alpha)n$,

$$\left\|\Sigma^{-1}\left(\frac{1}{|T|}\sum_{i \in T} x_i z_i\right)\right\| \leq C_2 K^2 \sigma \alpha \log^{2a}(1/\alpha) . \tag{29}$$

# H   Proof of Thm. H.3 on the analysis of Alg. 1

We provide our main theorem under the following sub-Weibull assumptions.

**Assumption H.1** (($\Sigma, \sigma^2, w^*, K, a$)-model). A multiset $S_{\text{good}} = \{(x_i \in \mathbb{R}^d, y_i \in \mathbb{R})\}_{i=1}^n$ of $n$ i.i.d. samples from a linear model $y_i = \langle x_i, w^*\rangle + z_i$, where the input vector $x_i$ is zero mean, $\mathbb{E}[x_i] = 0$, with a positive definite covariance $\Sigma := \mathbb{E}[x_i x_i^\top] \succ 0$, and the (input dependent) label noise $z_i$ is zero mean, $\mathbb{E}[z_i] = 0$, with variance $\sigma^2 := \mathbb{E}[z_i^2]$. We further assume $\mathbb{E}[x_i z_i] = 0$, which is equivalent to assuming that the true parameter $w^* = \Sigma^{-1}\mathbb{E}[y_i x_i]$. We assume that the marginal distribution of $x_i$ is $(K, a)$-sub-Weibull and that of $z_i$ is also $(K, a)$-sub-Weibull, as defined below.

Sub-Weibull distributions provide Gaussian-like tail bounds determining the resilience of the dataset in Lemma J.7, which our analysis critically relies on and whose necessity is justified in Sec. 3.4.

**Definition H.2** (sub-Weibull distribution [57] ). For some $K, a > 0$, we say a random vector $x \in \mathbb{R}^d$ is from a $(K, a)$-sub-Weibull distribution if for all $v \in \mathbb{R}^d$, $\mathbb{E}\left[\exp\left(\left(\frac{\langle v, x\rangle^2}{K^2 \mathbb{E}[\langle v, x\rangle^2]}\right)^{1/(2a)}\right)\right] \leq 2$.

**Theorem H.3.** *Alg. 1 is $(\varepsilon, \delta)$-DP. Under $(\Sigma, \sigma^2, w^*, K, a)$-model of Asmp. H.1 and $\alpha_{\text{corrupt}}$-corruption of Assumption 3.7 and for any failure probability $\zeta \in (0, 1)$ and target error rate $\alpha \geq \alpha_{\text{corrupt}}$. We further assume that the corruption level is bounded by $\alpha_{\text{corrupt}} \leq \bar{\alpha}$, where $\bar{\alpha}$ is a known positive constant satisfying $\bar{\alpha} \leq 1/10$, $72 C_2 K^2 \bar{\alpha} \log^{2a}(1/(6\bar{\alpha})) \log(\kappa) \leq 1/2$, and $2 C_2 K^2 \log^{2a}(1/(2\bar{\alpha})) \geq 1$ for the $(K, a)$-sub-Weibull distribution of interest and a positive constant $C_2$ defined in Lemma J.7 that only depends on $(K, a)$. If the sample size is large enough such that*

$$n = \tilde{O}\left(K^2 d \log^{2a+1}\left(\frac{1}{\zeta}\right) + \frac{d + \log(1/\zeta)}{\alpha^2} + \frac{K^2 d T^{1/2} \log(\frac{1}{\delta}) \log^a(\frac{1}{\zeta})}{\varepsilon \alpha}\right), \tag{30}$$

*with a large enough constant where $\tilde{O}$ hides poly-logarithmic terms in $d$, $n$, and $\kappa$, then the choices of a step size $\eta = 1/(C \lambda_{\max}(\Sigma))$ for any $C \geq 1.1$ and the number of iterations, $T = \tilde{\Theta}(\kappa \log(\|w^*\|))$ for a condition number of the covariance $\kappa := \lambda_{\max}(\Sigma)/\lambda_{\min}(\Sigma)$, ensures that, with probability $1 - \zeta$, Alg. 1 achieves*

$$\mathbb{E}_{\nu_1, \cdots, \nu_t \sim \mathcal{N}(0, \mathbf{I}_d)}\left[\|w_T - w^*\|_\Sigma^2\right] = \tilde{O}\left(K^4 \sigma^2 \alpha^2 \log^{4a}\left(\frac{1}{\alpha}\right)\right), \tag{31}$$

*where the expectation is taken over the noise added for DP, and $\tilde{\Theta}(\cdot)$ hides logarithmic terms in $K, \sigma, d, n, 1/\varepsilon, \log(1/\delta), 1/\alpha$, and $\kappa$.*

The main theorem builds upon the following lemma that analyzes a (stochastic) gradient descent method, where the randomness is from the DP noise we add and the analysis only relies on certain deterministic conditions on the dataset including resilienece and concentration. Thm. H.3 follows in a straightforward manner by collecting Thm. C.1, Lemma F.1, Lemma C.3, and Lemma H.4.

**Lemma H.4.** *Alg. 1 is $(\varepsilon, \delta)$-DP. Under Assumptions H.1 and 3.7 for any $\zeta \in (0, 1)$ and $\alpha \geq \alpha_{\text{corrupt}}$ satisfying $K^2 \alpha \log^{2a}(1/\alpha) \log(\kappa) \leq c$ for some universal constant $c > 0$, if distance threshold is small enough such that*

$$\theta_t \leq 3C_2^{1/2} K \log^a(1/(2\alpha)) \cdot (\|w^* - w_t\|_\Sigma + \sigma) , \tag{32}$$

*and large enough such that the number of clipped clean data points is no larger than $\alpha n$, at every round, the norm threshold is large enough such that*

$$\Theta \geq K \sqrt{\text{Tr}(\Sigma)} \log^a(n/\zeta) , \tag{33}$$

*and sample size is large enough such that*

$$n = O\left( K^2 d \log(d/\zeta) \log^{2a}(n/\zeta) + \frac{d + \log(1/\zeta)}{\alpha^2} + \frac{K^2 T^{1/2} d \log(T/\delta) \log^a(n/(\alpha\zeta))}{\varepsilon\alpha} \right) , \tag{34}$$

*with a large enough constant, then the choices of a step size, $\eta = 1/(C\lambda_{\max}(\Sigma))$ for some $C \geq 1.1$, and the number of iterations, $T = \tilde{\Theta}(\kappa \log(\|w^*\|))$, ensures that Alg. 1 outputs $w_T$ satisfying the following with probability $1 - \zeta$:*

$$\mathbb{E}_{\nu_1, \cdots, \nu_t \sim \mathcal{N}(0, \mathbf{I}_d)}[\|w_T - w^*\|_\Sigma^2] \lesssim K^4 \sigma^2 \log^2(\kappa)\alpha^2 \log^{4a}(1/\alpha) , \tag{35}$$

*where the expectation is taken over the noise added for DP and $\tilde{\Theta}(\cdot)$ hides logarithmic terms in $K, \sigma, d, n, 1/\varepsilon, \log(1/\delta), 1/\alpha$.*

*Proof of Lemma H.4.* We first prove a set of deterministic conditions on the clean dataset, which is sufficient for the analysis of the gradient descent.

**Step 1: Sufficient deterministic conditions on the clean dataset.** Let $S_{\text{good}}$ be the uncorrupted dataset for $S_3$ and $S_{\text{bad}}$ be the corrupted datapoints in $S_3$. Let $G := S_{\text{good}} \cap S_3 = S_3 \setminus S_{\text{bad}}$ denote the clean data that remains in the input dataset. Let $\lambda_{\max} = \|\Sigma\|_2$. Define $\hat{\Sigma} := (1/n) \sum_{i \in G} x_i x_i^\top$, $\hat{B} := \mathbf{I}_d - \eta\hat{\Sigma}$. Lemma J.4 implies that if $n = O(K^2 d \log(d/\zeta) \log^{2a}(n/\zeta))$, then

$$0.9\Sigma \preceq \hat{\Sigma} \preceq 1.1\Sigma . \tag{36}$$

We pick step size $\eta$ such that $\eta \leq 1/(1.1\lambda_{\max})$ to ensure that $\eta \leq 1/\|\hat{\Sigma}\|_2$. Since the covariates $\{x_i\}_{i \in S}$ are not corrupted, from Lemma J.3, we know with probability $1 - \zeta$, for all $i \in S_3$,

$$\|x_i\|^2 \leq K^2 \text{Tr}(\Sigma) \log^{2a}(n/\zeta) . \tag{37}$$

Lemma J.7 implies that if $n = O((d + \log(1/\zeta))/(\alpha^2))$, then there exists a universal constant $C_2$ such that $S_3$ is, following Definition J.6, with respect to $(w^*, \Sigma, \sigma)$,
$(\alpha_{\text{corrupt}}, \alpha, C_2 K^2 \alpha \log^{2a}(1/\alpha), C_2 K^2 \alpha \log^{2a}(1/\alpha), C_2 K^2 \alpha \log^{2a}(1/\alpha), C_2 K \alpha \log^a(1/\alpha))$-corrupt good. Such corrupt good sets have a sufficiently large, $1 - \alpha_{\text{corrupt}}$, fraction of points that satisfy a good property that we need: resilience. The rest of the proof is under Eq. (36), Eq. (37), and that $S_{\text{good}}$ is resilient.

**Step 2: Upper bounding the deterministic noise in the gradient.** In this step, we bound the deviation of the gradient from its mean. There are several sources of deviation: $(i)$ clipping, $(ii)$ adversarial corruptions, and $(iii)$ randomness of the data noise and privacy noise. We will show that deviations from all these sources can be controlled deterministically under the corrupt-goodness (i.e., resilience).

Let $\phi_t = (\sqrt{2\log(1.25/\delta_0)}\Theta\theta_t)/(\varepsilon_0 n)$, which ensures that we add enough noise to guarantee $(\varepsilon_0, \delta_0)$-DP for each step of gradient descent. This follows from the standard Gaussian mechanism in

Lemma B.1 and the fact that each gradient is clipped to the norm of $\Theta\theta_t$, resulting in a DP sensitivity of $\Theta\theta_t/n$. The fact that this sensitivity scales as $1/n$ is one of the main reasons for the performance gain we get over [81] that uses a minimatch of size $n/\kappa$ with sensitivity scaling as $\kappa/n$. Define $g_i^{(t)} := x_i(x_i^\top w_t - y_i)$. For $i \in S_{\text{good}}$, we know $y_i = x_i^\top w^* + z_i$. Let $\tilde{g}_i^{(t)} = \text{clip}_\Theta(x_i)\text{clip}_{\theta_t}(x_i^\top w_t - y_i)$. Note that under Eq. (37), $\text{clip}_\Theta(x_i) = x_i$ for all $i \in S_3$.

From Alg. 1, we can write one-step update rule as follows:

$$
\begin{aligned}
&w_{t+1} - w^* \\
=&w_t - \eta\left(\frac{1}{n}\sum_{i\in S}\tilde{g}_i^{(t)} + \phi_t\nu_t\right) - w^* \\
=&\left(\mathbf{I} - \frac{\eta}{n}\sum_{i\in G}x_ix_i^\top\right)(w_t - w^*) + \frac{\eta}{n}\sum_{i\in G}x_iz_i + \frac{\eta}{n}\sum_{i\in G}(g_i^{(t)} - \tilde{g}_i^{(t)}) - \eta\phi_t\nu_t - \frac{\eta}{n}\sum_{i\in S_{\text{bad}}}\tilde{g}_i^{(t)}
\end{aligned}
$$

$$(38)$$

Let $E_t := \{i \in G : \theta_t \le |x_i^\top w_t - y_i|\}$ be the set of clipped clean data points such that $\sum_{i\in G}(g_i^{(t)} - \tilde{g}_i^{(t)}) = \sum_{i\in E_t}(g_i^{(t)} - \tilde{g}_i^{(t)})$. We define $\hat{v} := (1/n)\sum_{i\in G}x_iz_i$, $u_t^{(1)} := (1/n)\sum_{i\in E_t}x_ix_i^\top(w_t - w^*)$, $u_t^{(2)} := (1/n)\sum_{i\in E_t}-x_iz_i$, and $u_t^{(3)} := (1/n)\sum_{i\in S_{\text{bad}}\cup E_t}\tilde{g}_i^{(t)}$.

We can further write the update rule as:

$$w_{t+1} - w^* = \hat{B}(w_t - w^*) + \eta\hat{v} + \eta u_{t-1}^{(1)} + \eta u_{t-1}^{(2)} - \eta\phi_t\nu_t - \eta u_{t-1}^{(3)}. \tag{39}$$

We bound each term one-by-one. Since $G \subset S_{\text{good}}$ and $|G| = (1 - \alpha_{\text{corrupt}})n$, using the resilience property in Eq. (5), we know

$$
\begin{aligned}
\|\Sigma^{-1/2}\hat{v}\| &= (1 - \alpha_{\text{corrupt}})\max_{\|v\|=1}\Sigma^{-1/2}\left\langle v, \frac{1}{(1-\alpha_{\text{corrupt}})n}\sum_{i\in G}x_iz_i\right\rangle \\
&\le (1 - \alpha_{\text{corrupt}})C_2K^2\alpha\log^{2a}(1/\alpha)\sigma \tag{40} \\
&\le C_2K^2\alpha\log^{2a}(1/\alpha)\sigma. \tag{41}
\end{aligned}
$$

Let $\tilde{\alpha} = |E_t|/n$. By assumption, we know $\tilde{\alpha} \le \alpha$ (which holds for the given dataset due to Lemma C.3), and

$$\|\Sigma^{-1/2}u_t^{(1)}\| = \|\Sigma^{-1/2}\frac{1}{n}\sum_{i\in E_t}x_ix_i^\top(w_t - w^*)\|.$$

From Corollary J.8, we know

$$\left| \|\Sigma^{-1/2}\frac{1}{|E_t|}\sum_{i\in E_t} x_i x_i^\top (w_t - w^*)\| - \|w_t - w^*\|_\Sigma \right|$$

$$= \left| \max_{u:\|u\|=1} \frac{1}{|E_t|}\sum_{i\in E_t} u^\top \Sigma^{-1/2} x_i x_i^\top (w_t - w^*)\| - \max_{v:\|v\|=1} v^\top \Sigma^{1/2}(w_t - w^*) \right|$$

$$\leq \max_{u:\|u\|=1} \left| \frac{1}{|E_t|}\sum_{i\in E_t} u^\top \Sigma^{-1/2} x_i x_i^\top \Sigma^{-1/2}\Sigma^{1/2}(w_t - w^*)\| - u^\top \Sigma^{1/2}(w_t - w^*) \right|$$

$$\leq \max_{u:\|u\|=1} \left| \frac{1}{|E_t|}\sum_{i\in E_t} u^\top \left( \Sigma^{-1/2} x_i x_i^\top \Sigma^{-1/2} - \mathbf{I}_d \right) \Sigma^{1/2}(w_t - w^*)\| \right|$$

$$= \left\| \frac{1}{|E_t|}\sum_{i\in E_t} \left( \Sigma^{-1/2} x_i x_i^\top \Sigma^{-1/2} - \mathbf{I}_d \right) \Sigma^{1/2}(w_t - w^*) \right\|$$

$$\leq \left\| \frac{1}{|E_t|}\sum_{i\in E_t} \left( \Sigma^{-1/2} x_i x_i^\top \Sigma^{-1/2} - \mathbf{I}_d \right) \right\| \cdot \left\| \Sigma^{1/2}(w_t - w^*) \right\|$$

$$\leq \frac{2-\tilde{\alpha}}{\tilde{\alpha}} C_2 K^2 \alpha \log^{2a}(1/\alpha) \|w_t - w^*\|_\Sigma .$$

This implies that

$$\|\Sigma^{-1/2} u_t^{(1)}\| \leq \|\Sigma^{-1/2}\frac{1}{n}\sum_{i\in E} x_i x_i^\top (w_t - w^*)\|$$

$$\leq \left( \tilde{\alpha} + 2C_2 K^2 \alpha \log^{2a}(1/\alpha) \right) \|w_t - w^*\|_\Sigma$$

$$\leq 3C_2 K^2 \alpha \log^{2a}(1/\alpha) \|w_t - w^*\|_\Sigma , \tag{42}$$

where the last inequality follows from the fact that $\tilde{\alpha} \leq \alpha$ and our assumption that $C_2 K^2 \log^{2a}(1/\bar{\alpha}) \geq 1$ from Asmp. 3.7. Similarly, we use resilience property in Eq. (5) instead of Eq. (6), we can show that

$$\|\Sigma^{-1/2} u_t^{(2)}\| \leq 3C_2 K^2 \alpha \log^{2a}(1/\alpha)\sigma . \tag{43}$$

Next, we consider $u_t^{(3)}$. Since $|S_{\text{bad}}| \leq \alpha_{\text{corrupt}} n$ and $|E_t| \leq \alpha n$, using Eq. (8) and Corollary J.8, we have

$$\|\Sigma^{-1/2} u_t^{(3)}\| = \max_{v:\|v\|=1} \frac{1}{n}\sum_{i\in S_{\text{bad}}\cup E_t} v^\top \Sigma^{-1/2} x_i \text{clip}_{\theta_t}(x_i^\top w_t - y_i)$$

$$\leq 2C_2 K \alpha \log^a(1/\alpha)\theta_t$$

$$\leq 6C_2^{1.5} K^2 \alpha \log^{2a}(1/\alpha)(\|w_t - w^*\|_\Sigma + \sigma) . \tag{44}$$

Now we use Eq. (41), Eq. (42), Eq. (43) and Eq. (44) to bound the final error from update rule in Eq. (39).

**Step 3: Analysis of the $t$-steps recurrence relation.** We have controlled the deterministic noise in the last step. In this step, we will upper bound the noise introduced by the Gaussian noise for the purpose of privacy, and show the expected distance to optimum decrease every step.

We want to emphasize that most of our technical contribution is in the convergence analysis (Step 3 and Step 4). More precisely, naive linear regression analysis can only show a suboptimal error rate of $\|\hat{w} - w^\star\|_\Sigma = \tilde{O}(\kappa\alpha\sigma)$ with sample size $n = \tilde{O}(d/\alpha^2 + \kappa^{1/2}d/(\varepsilon\alpha))$. Define $u_t = (\hat{v} + u_t^{(1)} + u_t^{(2)} - u_t^{(3)})$. This follows from Eq. (39):

$$w_{t+1} - w^* = \hat{B}(w_t - w^*) + \eta u_t - \eta\phi_t\nu_t \tag{45}$$

$$= (\mathbf{I}_d - \eta\hat{\Sigma})(w_t - w^*) + \eta u_t - \eta\phi_t\nu_t . \tag{46}$$

From Eq. (42), Eq. (43) and Eq. (44), it follows that

$$\|w_{t+1} - w^*\|_\Sigma \leq (1 - \frac{1}{\kappa})\|w_t - w^*\|_\Sigma + \alpha(\sigma + \|w_t - w^*\|_\Sigma)$$

where we omitted constants for simplicity, which after $T = \tilde{O}(\kappa)$ iterations achieves a *sub-optimal* error rate $\|w_T - w^*\|_\Sigma = \tilde{O}(\kappa\alpha\sigma)$.

One attempt to get around it is to take the Euclidean norm instead, which gives, after some calculations,

$$\mathbb{E}[\|w_{t+1} - w^*\|^2] \leq \mathbb{E}[\|w_t - w^*\|^2] - \eta\Big(\|w_t - w^*\|_\Sigma^2 - \alpha^2\sigma^2\Big) .$$

This implies that $\mathbb{E}[\|w_{t+1} - w^*\|^2]$ strictly decreases as long as $\|w_t - w^*\|_\Sigma^2 > C\alpha^2\sigma^2$, which is the desired statistical error level we are targeting. With this analysis, we can show that in $T = \tilde{O}(\kappa)$ iterations, there exists at least one model $w_t$ that achieves $\mathbb{E}[\|w_t - w^*\|_\Sigma^2] = \tilde{O}(\alpha^2\sigma^2)$ among all the intermediate models we have seen.

However, the problem is that under differential privacy, there is no way we could select this good model $w_t$ among $T$ models that we have, as privacy-preserving techniques for model selection are not accurate enough to achieve the desired level of accuracy. Hence, we came up with the following novel analysis that does not suffer from such issues.

We can rewrite Eq. (39) or Eq. (45) as

$$w_{t+1} - w^* = \hat{B}(w_t - w^*) + \eta u_t - \eta\phi_t\nu_t \tag{47}$$

$$= \hat{B}^{t+1}(w_0 - w^*) + \eta\sum_{i=0}^t \hat{B}^i u_{t-i} - \eta\sum_{i=0}^t \phi_{t-i}\hat{B}^i\nu_{t-i} . \tag{48}$$

Taking expectations of $\hat{\Sigma}$-norm square with respect to $\nu_1, \cdots, \nu_t$, we have

$$\mathbb{E}_{\nu_1,\ldots,\nu_t \sim \mathcal{N}(0,\mathbf{I}_d)}\|w_{t+1} - w^*\|_{\hat{\Sigma}}^2 \tag{49}$$

$$\leq 2\|\hat{B}^{t+1}(w_0 - w^*)\|_{\hat{\Sigma}}^2 + 2\mathbb{E}[\|\eta\sum_{i=0}^t \hat{B}^i u_{t-i}\|_{\hat{\Sigma}}^2] + \eta^2\sum_{i=0}^t \mathrm{Tr}(\hat{B}^{2i}\hat{\Sigma})\mathbb{E}[\phi_{t-i}^2] \tag{50}$$

$$\leq 2\|\hat{B}^{t+1}(w_0 - w^*)\|_{\hat{\Sigma}}^2 + 2\eta^2\mathbb{E}[\sum_{i=0}^t\sum_{j=0}^t \|\hat{B}^i u_{t-i}\|_{\hat{\Sigma}}\|\hat{B}^j u_{t-j}\|_{\hat{\Sigma}}] \tag{51}$$

$$+ \eta^2\sum_{i=0}^t \mathrm{Tr}(\hat{B}^{2i}\hat{\Sigma})\mathbb{E}[\phi_{t-i}^2] , \tag{52}$$

where at the second step we used the fact that $\nu_1, \nu_2, \cdots, \nu_t$ are independent isotropic Gaussian.

Note that

$$\begin{aligned}
\eta\|\hat{B}^i u_{t-i}\|_{\hat{\Sigma}} &= \eta\|\hat{\Sigma}^{1/2}\hat{B}^i\hat{\Sigma}^{1/2}\hat{\Sigma}^{-1/2}u_{t-i}\| \\
&\leq \eta\|\hat{\Sigma}^{1/2}\hat{B}^i\hat{\Sigma}^{1/2}\|_2 \cdot \|\hat{\Sigma}^{-1/2}u_{t-i}\| \\
&\leq \eta\|\hat{\Sigma}^{1/2}\hat{B}^i\hat{\Sigma}^{1/2}\|_2\,\hat{\rho}(\alpha)\,(\|w_{t-i} - w^*\|_{\hat{\Sigma}} + \sigma) \\
&\leq \frac{1}{i+1}\hat{\rho}(\alpha)\,(\|w_{t-i} - w^*\|_{\hat{\Sigma}} + \sigma) ,
\end{aligned}$$

where $\hat{\rho}(\alpha) = 1.1(6C_2 + 6C_2^{1.5})K^2\alpha\log^{2a}(1/\alpha)$, and the second inequality follows from Eq. (42), Eq. (43), Eq. (44) and the deterministic condition in Eq. (36). Note that the last inequality is true because $\eta \leq 1/(1.1\lambda_{\max})$ and $\|\hat{\Sigma}^{1/2}\hat{B}^i\hat{\Sigma}^{1/2}\|_2 \leq \|\mathbf{I}_d - \eta\hat{\Sigma}\|_2^i\|\hat{\Sigma}\|_2 \leq \lambda_{\max}/(i+1) .$

This implies

$$\mathbb{E}[\eta^2 \sum_{i=0}^{t} \sum_{j=0}^{t} \|\hat{B}^i u_{t-i}\|_{\hat{\Sigma}} \|\hat{B}^j u_{t-j}\|_{\hat{\Sigma}}] \tag{53}$$

$$\leq 4 \mathbb{E}[\sum_{i=0}^{t} \sum_{j=0}^{t} \frac{\hat{\rho}(\alpha)^2}{(i+1)(j+1)} (\mathbb{E}[\|w_{t-i} - w^*\|_{\hat{\Sigma}}^2] + \mathbb{E}[\|w_{t-j} - w^*\|_{\hat{\Sigma}}^2] + \sigma^2) \tag{54}$$

$$\leq 8 (\sum_{i=0}^{t} \frac{1}{i+1})^2 \hat{\rho}(\alpha)^2 (\max_i \mathbb{E}[\|w_{t-i} - w^*\|_{\hat{\Sigma}}^2] + \sigma^2) \tag{55}$$

$$\leq 8 (\log t)^2 \hat{\rho}(\alpha)^2 (\max_i \mathbb{E}[\|w_{t-i} - w^*\|_{\hat{\Sigma}}^2] + \sigma^2) , \tag{56}$$

Then,

$$\|\hat{B}^{t+1}(w_0 - w^*)\|_{\hat{\Sigma}}^2 = \|\hat{\Sigma}^{1/2} \hat{B}^{t+1} \hat{\Sigma}^{-1/2} \hat{\Sigma}^{1/2} (w_0 - w^*)\|^2$$
$$\leq (1 - \frac{1}{\kappa})^{2(t+1)} \|w_0 - w^*\|_{\hat{\Sigma}}^2 \leq e^{-2(t+1)/\kappa} \|w_0 - w^*\|_{\hat{\Sigma}}^2 ,$$

and for $n \gtrsim (1/\varepsilon)\sqrt{\kappa d \log(1/\delta)/\alpha}$,

$$\eta^2 \sum_{i=0}^{t} \text{Tr}(\hat{B}^{2i} \hat{\Sigma}) \mathbb{E}[\phi_{t-i}^2] \tag{57}$$

$$\leq \eta^2 \sum_{i=0}^{t} \|\mathbf{I}_d - \eta\hat{\Sigma}\|_2^{2i} \|\hat{\Sigma}\|_2 \cdot \frac{2\log(1.25/\delta_0) K^2 \text{Tr}(\Sigma) \log^{2a}(n/\zeta_0) C_2 K^2 \log^{2a}(1/(2\alpha))(\mathbb{E}[\|w_{t-i} - w^*\|_{\Sigma}^2] + \sigma^2)}{\varepsilon_0^2 n^2} \tag{58}$$

$$\leq 4 \sum_{i=0}^{t} (\frac{1}{i+1})^2 \hat{\rho}(\alpha)^2 (\mathbb{E}[\|w_{t-i} - w^*\|_{\hat{\Sigma}}^2] + \sigma^2) . \tag{59}$$

We have

$$\mathbb{E}_{\nu_1,\dots,\nu_t \sim \mathcal{N}(0,\mathbf{I}_d)}[\|w_{t+1} - w^*\|_{\hat{\Sigma}}^2] \leq 2e^{-2(t+1)/\kappa} \|w_0 - w^*\|_{\hat{\Sigma}}^2 + 20(\log t)^2 \hat{\rho}(\alpha)^2 (\max_{i \in [t]} \mathbb{E}[\|w_{t-i} - w^*\|_{\hat{\Sigma}}^2] + \sigma^2) .$$

Note that this also implies that

$$\mathbb{E}[\|(w_{t'+t} - w^*)\|_{\hat{\Sigma}}^2 | w_{t'}] \leq 2e^{-2t/\kappa} \|w_{t'} - w^*\|_{\hat{\Sigma}}^2 + 20\hat{\rho}(\alpha)^2 \sum_{i=0}^{t-1} (\frac{1}{i+1})^2 (\mathbb{E}[\|w_{t'+t-i} - w^*\|_{\hat{\Sigma}}^2 | w_{t'}] + \sigma^2) , \tag{60}$$

which implies

$$\mathbb{E}[\|(w_{t'+t} - w^*)\|_{\hat{\Sigma}}^2] \leq 2e^{-2t/\kappa} \mathbb{E}[\|w_{t'} - w^*\|_{\hat{\Sigma}}^2] + 20\hat{\rho}(\alpha)^2 \sum_{i=0}^{t-1} (\frac{1}{i+1})^2 (\mathbb{E}[\|w_{t'+t-i} - w^*\|_{\hat{\Sigma}}^2] + \sigma^2) \tag{61}$$

$$\leq 2e^{-2t/\kappa} \mathbb{E}[\|w_{t'} - w^*\|_{\hat{\Sigma}}^2] + 20(\log t)^2 \hat{\rho}(\alpha)^2 (\max_{i \in [t]} \mathbb{E}[\|w_{t'+t-i} - w^*\|_{\hat{\Sigma}}^2] + \sigma^2) \tag{62}$$

**Step 4: End-to-end analysis of the convergence.** In the last step, we shown that the amount of estimation error decrease depends on the estimation error of the previous $t$ steps. In order for the estimation error to decrease by a constant factor, we will take $t = \kappa$. Roughly speaking, we will prove that for every $\kappa$ steps, the estimation error will decrease by a constant factor, if it is much larger than $O((\log \kappa)^2 \hat{\rho}(\alpha)^2 \sigma^2)$. This implies we will reach $O((\log \kappa)^2 \hat{\rho}(\alpha)^2 \sigma^2)$ error with in $\tilde{O}(\kappa)$ steps.

For any integer $s \geq 0$, as long as $\max_{i \in [(s-1)\kappa+1, s\kappa]} \mathbb{E}[\|w_i - w^*\|_{\hat{\Sigma}}^2] \geq 2(\log \kappa)^2 \hat{\rho}(\alpha)^2 \sigma^2$,

$$\max_{i \in [s\kappa+1, (s+1)\kappa]} \mathbb{E}[\|w_i - w^*\|_{\hat{\Sigma}}^2] \leq (\frac{1}{e^2} + (\log \kappa)^2 \hat{\rho}(\alpha)^2) \max_{i \in [(s-1)\kappa+1, s\kappa]} \mathbb{E}[\|w_i - w^*\|_{\hat{\Sigma}}^2] + (\log 2\kappa)^2 \hat{\rho}(\alpha)^2 \sigma^2 . \tag{63}$$

Assuming $\hat{\rho}(\alpha)^2 (\log \kappa)^2 \leq 1/2 - 1/e^2$, the maximum expected error in a length $\kappa$ sequence decrease by a factor of $1/2$ every time.

Now we bound the maximum expected error in the first length $\kappa$ sequence: $\max_{i \in [0, \kappa-1]} \mathbb{E}[\|w_i - w^*\|_{\hat{\Sigma}}^2]$. Since

$$\mathbb{E}[\|w_i - w^*\|_{\hat{\Sigma}}^2] \leq e^{-2i/\kappa} \|w_0 - w^*\|_{\hat{\Sigma}}^2 + (\log i)^2 \hat{\rho}(\alpha)^2 \max_{j \in [0, i-1]} \mathbb{E}[\|w_j - w^*\|_{\hat{\Sigma}}^2] + (\log i)^2 \hat{\rho}(\alpha)^2 \sigma^2 .$$

As a function of $i$, $\max_{j \in [0, i-1]} \mathbb{E}[\|w_j - w^*\|_{\hat{\Sigma}}^2]$ only increase when it is smaller than

$$\frac{1}{1 - (\log i)^2 \hat{\rho}(\alpha)^2} (\|w_0 - w^*\|_{\hat{\Sigma}}^2 + (\log i)^2 \hat{\rho}(\alpha)^2 \sigma^2) .$$

Thus we conclude

$$\max_{i \in [0, \kappa-1]} \mathbb{E}[\|w_i - w^*\|_{\hat{\Sigma}}^2] \leq \frac{1}{1 - (\log \kappa)^2 \hat{\rho}(\alpha^2)} (\|w_0 - w^*\|_{\hat{\Sigma}}^2 + (\log \kappa)^2 \hat{\rho}(\alpha^2) \sigma^2)$$

$s = \log(\|w^*\| / (\hat{\rho}(\alpha)\sigma))$ will give us

$$\mathbb{E}[\|w_{s\kappa+1} - w^*\|_{\hat{\Sigma}}^2] \leq (\log \kappa)^2 \hat{\rho}(\alpha)^2 \sigma^2 .$$

$\square$

# I  Lower bounds

## I.1  Proof of Proposition 3.9 for label corruption lower bounds

We first prove the following lemma.

**Lemma I.1.** *Consider an $\alpha$ label-corrupted dataset $S = \{(x_i, y_i)\}_{i=1}^n$ with $\alpha < 1/2$, that is generated from either $x_i \sim \mathcal{N}(0,1), y_i \sim \mathcal{N}(0,1)$ or $x_i \sim \mathcal{N}(0,1), z_i \sim \mathcal{N}(0, 1-\alpha^2), y_i = \alpha x_i + z_i$. It is impossible to distinguish the two hypotheses with probability larger than $1/2$.*

In the first case,

$$(x_i, y_i) \sim \mathcal{P}_1 = \mathcal{N}(0, \begin{bmatrix} 1 & 0 \\ 0 & 1 \end{bmatrix}).$$

In the second case,

$$(x_i, y_i) \sim \mathcal{P}_2 = \mathcal{N}(0, \begin{bmatrix} 1 & \alpha \\ \alpha & 1 \end{bmatrix}).$$

By simple calculation, it holds that $D_{KL}(\mathcal{P}_1 \| \mathcal{P}_2) = -\frac{1}{2} \log(1 - \alpha^2) \leq \alpha^2/2$ for all $\alpha < 1/2$. Then, Pinsker's inequality implies that $D_{TV}(\mathcal{P}_1 \| \mathcal{P}_2) \leq \alpha/2$. Since the covariate $x_i$ follows from the same distribution in the two cases, and the total variation distance between the two cases is less than $\alpha/2$. This means there is an label corruption adversary that change $\alpha/2$ fraction of $y_i$'s in $\mathcal{P}_1$ to make it identical to $\mathcal{P}_2$. Therefore, no algorithm can distinguish the two cases with probability better than $1/2$ under $\alpha$ fraction of label corruption.

Since $\Sigma = 1, \sigma^2 \in [3/4, 1]$, the first case above has $w^* = 0$, and the second case has $w^* = \alpha$, this implies that no algorithm is able to achieve $\mathbb{E}[\|\hat{w} - w^*\|_\Sigma] < \sigma\alpha$ for all instances with $\|w^*\| \leq 1$ under $\alpha$ fraction of label corruption.

# J  Technical Lemmas

**Lemma J.1** (Hanson-Wright inequality for subWeibull distributions [69]). *Let $S = \{x_i \in \mathbb{R}^d\}_{i=1}^n$ be a dataset consist of i.i.d. samples from $(K, a)$-subWeibull distributions, then*

$$\mathbb{P}\left(\left|\frac{1}{n} \sum_{i=1}^n \|x_i\|^2 - \text{Tr}(\Sigma)\right| \geq t\right) \leq 2 \exp\left(-\min\left\{\frac{nt^2}{K^4 (\text{Tr}(\Sigma))^2}, \left(\frac{nt}{K^2 \text{Tr}(\Sigma)}\right)^{\frac{1}{2a}}\right\}\right) . \quad (64)$$

**Lemma J.2.** *Let $Y \sim \text{Lap}(b)$. Then for all $h > 0$, we have $\mathbb{P}(|Y| \geq hb) = e^{-h}$.*

**Lemma J.3.** *If $x \in \mathbb{R}^d$ is $(K, a)$-subWeibull for some $a \in [1/2, \infty)$. Then*

* *for any fixed $v \in \mathbb{R}^d$, with probability $1 - \zeta$,*

$$\langle x, v \rangle^2 \leq K^2 v^\top \Sigma v \log^{2a}(1/\zeta) . \tag{65}$$

* *with probability $1 - \zeta$,*

$$\|x\|^2 \leq K^2 \operatorname{Tr}(\Sigma) \log^{2a}(1/\zeta) . \tag{66}$$

We provide a proof in App. J.1.1.

**Lemma J.4.** *Dataset $S = \{x_i \in \mathbb{R}^d\}_{i=1}^n$ consists i.i.d. samples from a zero mean distribution $\mathcal{D}$. Suppose $\mathcal{D}$ is $(K, a)$-subWeibull. Define $\Sigma = \mathbb{E}_{x \sim \mathcal{D}}[xx^\top]$. Then there exists a constant $c_1 > 0$ such that with probability $1 - \zeta$,*

$$\left\| \frac{1}{n} \sum_{i=1}^n x_i x_i^\top - \Sigma \right\| \leq c_1 \left( \frac{K^2 d \log(d/\zeta) \log^{2a}(n/\zeta)}{n} + \sqrt{\frac{K^2 d \log(d/\delta) \log^{2a}(n/\zeta)}{n}} \right) \|\Sigma\|_2 . \tag{67}$$

**Lemma J.5** (Lemma F.1 from [59])**.** *Let $x \in \mathbb{R}^d \sim \mathcal{N}(0, \Sigma)$. Then there exists universal constant $C_6$ such that with probability $1 - \zeta$,*

$$\|x\|^2 \leq C \operatorname{Tr}(\Sigma) \log(1/\zeta) . \tag{68}$$

**Definition J.6** (Corrupt good set)**.** We say a dataset $S$ is $(\alpha_{\text{corrupt}}, \alpha, \rho_1, \rho_2, \rho_3, \rho_4)$-corrupt good with respect to $(w^*, \Sigma, \sigma)$ if it is $\alpha_{\text{corrupt}}$-corruption of an $(\alpha, \rho_1, \rho_2, \rho_3, \rho_4)$-resilient dataset $S_{\text{good}}$.

**Lemma J.7.** *Under Assumptions H.1 and 3.7, there exists positive constants $c_1$ and $C_2$ such that if $n \geq c_1((d + \log(1/\zeta))/\alpha^2$, then with probability $1 - \zeta$, $S_{\text{good}}$ is, with respect to $(w^*, \Sigma, \sigma)$, $(\alpha, C_2 K^2 \alpha \log^{2a}(1/\alpha), C_2 K^2 \alpha \log^{2a}(1/\alpha), C_2 K^2 \alpha \log^{2a}(1/\alpha), C_2 K \alpha \log^a(1/\alpha))$-resilient.*

We provide a proof in App. G.

**Corollary J.8** (Lemma 10 from [73] and Lemma 25 from [61])**.** *For a $(\alpha, \rho_1, \rho_2, \rho_3, \rho_4)$-resilient set $S$ with respect to $(w^*, \Sigma, \gamma)$ and any $0 \leq \tilde{\alpha} \leq \alpha$, the following holds for any subset $T \subset S$ of size at least $\tilde{\alpha} n$ and for any unit vector $v \in \mathbb{R}^d$:*

$$\left| \frac{1}{|T|} \sum_{(x_i, y_i) \in T} \langle v, x_i \rangle (y_i - x_i^\top w^*) \right| \leq \frac{2 - \tilde{\alpha}}{\tilde{\alpha}} \rho_1 \sqrt{v^\top \Sigma v} \, \sigma , \tag{69}$$

$$\left| \frac{1}{|T|} \sum_{x_i \in T} \langle v, x_i \rangle^2 - v^\top \Sigma v \right| \leq \frac{2 - \tilde{\alpha}}{\tilde{\alpha}} \rho_2 v^\top \Sigma v , \tag{70}$$

$$\left| \frac{1}{|T|} \sum_{(x_i, y_i) \in T} (y_i - x_i^\top w^*)^2 - \sigma^2 \right| \leq \frac{2 - \tilde{\alpha}}{\tilde{\alpha}} \rho_3 \sigma^2 , \quad \text{and} \tag{71}$$

$$\left| \frac{1}{|T|} \sum_{x_i \in T} \langle v, x_i \rangle \right| \leq \frac{2 - \tilde{\alpha}}{\tilde{\alpha}} \rho_4 \sqrt{v^\top \Sigma v} . \tag{72}$$

## J.1 Proof of technical lemmas

### J.1.1 Proof of Lemma J.3

Using Markov inequality,

$$\mathbb{P}\left( \langle v, x \rangle^2 \geq t^2 \right) = \mathbb{P}\left( e^{\langle v, x \rangle^{1/a}} \geq e^{t^{1/a}} \right) \tag{73}$$

$$\leq e^{-t^{1/a}} \mathbb{E}[e^{\langle v, x \rangle^{1/a}}] \tag{74}$$

$$\leq e^{-t^{1/a}} e^{K(\mathbb{E}[\langle v, x \rangle^2])^{1/(2a)}} \tag{75}$$

$$= 2 \exp\left( -\left( \frac{t^2}{K^2 \mathbb{E}[\langle v, x \rangle^2]} \right)^{1/(2a)} \right) . \tag{76}$$

This implies for any fixed $v$, with probability $1 - \zeta$,

$$\langle x, v \rangle^2 \le K^2 v^\top \mathbb{E}[xx^\top] v \log^{2a}(1/\zeta) . \tag{77}$$

For $j$-th coordinate, let $v = e_j$ where $j \in [d]$. Definition H.2 implies

$$\mathbb{E}\left[\exp\left(\left(\frac{x_j^2}{K^2 \operatorname{Tr}(\Sigma)}\right)^{1/(2a)}\right)\right] \le \mathbb{E}\left[\exp\left(\left(\frac{x_j^2}{K^2 \Sigma_{jj}}\right)^{1/(2a)}\right)\right] \le 2 . \tag{78}$$

Note that $f(x) = x^\alpha$ is concave function for $\alpha \le 1$ and $x > 0$. Then $(a_1 + \cdots a_k)^\alpha \le a_1^\alpha + \cdots a_k^\alpha$ holds for any positive numbers $a_1, \cdots, a_k > 0$. By our assumption that $1/(2a) \le 1.$ , we have

$$\mathbb{E}[\exp\left(\left(\frac{\|x\|^2}{K^2 \operatorname{Tr}(\Sigma)}\right)^{1/(2a)}\right)] = \mathbb{E}[\exp\left(\left(\frac{x_1^2 + x_2^2 + \cdots + x_d^2}{K^2 \operatorname{Tr}(\Sigma)}\right)^{1/(2a)}\right)] \tag{79}$$

$$\le \mathbb{E}[\prod_{j=1}^d \exp\left(\left(\frac{x_j^2}{K^2 \operatorname{Tr}(\Sigma)}\right)^{1/(2a)}\right)] \tag{80}$$

$$\le \left(\frac{\sum_{j=1}^d \mathbb{E}[\exp\left(\left(\frac{x_j^2}{K^2 \operatorname{Tr}(\Sigma)}\right)^{1/(2a)}\right)]}{d}\right)^d \tag{81}$$

$$\le 2 . \tag{82}$$

By Markov inequality,

$$\mathbb{P}\left(\|x\| \ge t\right) = \mathbb{P}\left(e^{\|x\|^{1/a}} \ge e^{t^{1/a}}\right) \tag{83}$$

$$\le e^{-t^{1/a}} \mathbb{E}[e^{\|x\|^{1/a}}] \tag{84}$$

$$\le \exp\left(-\left(\frac{t^2}{K^2 \operatorname{Tr}(\Sigma)}\right)^{1/(2a)}\right) . \tag{85}$$

This implies with probability $1 - \zeta$,

$$\|x\|^2 \le K^2 \operatorname{Tr}(\Sigma) \log^{2a}(1/\zeta) . \tag{86}$$

## K  Experiments

### K.1  DP Linear Regression

Experimental results for $\epsilon = 0.1$ can be found in Figure 2. The observations are similar to the $\epsilon = 1$ case. In particular, DP-SSP has poor performance when $\sigma$ is small. In other settings, DP-SSP has better performance than DP-RoBGD.

### K.2  DP Robust Linear Regression

We now illustrate the robustness of our algorithm. We consider the same experimental setup as in Sec. 4 and randomly corrupt $\alpha$ fraction of the response variables by setting them to 1000. Figure 3 presents the results from this experiment. It can be seen that none of the baselines are robust to adversarial corruptions. They can be made arbitrarily bad by increasing the magnitude of corruptions. In contrast, DP-RoBGD is able to handle the corruptions well.

### K.3  Stronger adversary for DP Robust Linear Regression

In this section, we consider a stronger adversary for DP-RoBGD than the one considered in Sec. 4. Recall, for the adversary model considered in Sec. 4, DP-RoBGD was able to consistently estimate

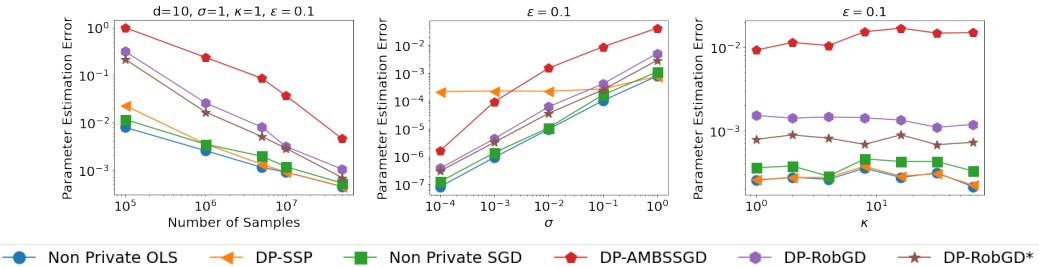

Figure 2: Performance of various techniques on DP linear regression. $d = 10$ in all the experiments. $n = 10^7, \kappa = 1$ in the $2^{nd}$ experiment. $n = 10^7, \sigma = 1$ in the $3^{rd}$ experiment.

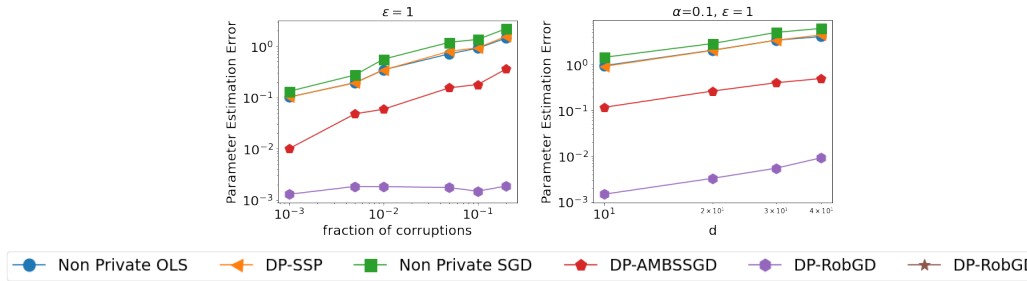

Figure 3: Non-robustness of existing techniques to adversarial corruptions. $n = 10^7, \sigma = 1$ in both experiments.

the parameter $w^*$ (*i.e.,* the parameter recovery error goes down to 0 as $n \to \infty$). This is because the algorithm was able to easily identify the corruptions and ignore the corresponding points while performing gradient descent. We now construct a different instance where the corruptions are hard to identify. Consequently, DP-RoBGD can no longer be consistent against the adversary. This hard instance is inspired by the lower bound in [11] (see Theorem 6.1 of [11]). This is a 2 dimensional problem where the first covariate is sampled uniformly from $[-1, 1]$. The second covariate, which is uncorrelated from the first, is sampled from a distribution with the following pdf

$$p(x^{(2)}) = \begin{cases} \frac{\alpha}{2} & \text{if } x^{(2)} \in \{-1, 1\} \\ \frac{1-\alpha}{2\alpha\sigma} & \text{if } x^{(2)} \in [-\sigma, \sigma] \\ 0 & \text{otherwise} \end{cases}.$$

We set $\sigma = 0.1$ in our experiments. The noise $z_i$ is sampled uniformly from $[-\sigma, \sigma]$. We consider two possible parameter vectors $w^* = (1, 1)$ and $w^* = (1, -1)$. It can be shown that the total variation (TV) distance between these problem instances (each parameter vector corresponds to one problem instance) is $\Theta(\alpha)$ [11]. What this implies is that, one can corrupt at most $\alpha$ fraction of the response variables and convert one problem instance into another. Since the distance (in $\Sigma$ norm) between the two parameter vectors is $\Omega(\alpha\sigma)$, any algorithm will suffer an error of $\Omega(\alpha\sigma)$.

We generate $10^7$ samples from this problem instance and add corruptions that convert one problem instance to the other. Figure 4 presents the results from this experiment. It can be seen that our algorithm works as expected. In particular, it is not consistent in this setting. Moreover, the parameter recovery error increases with the fraction of corruptions.

## L Heavy-tailed noise

We study the heavy-tailed regression settings where the label noise $z_i$ is hypercontractive, which is common in robust linear regression literature [54, 61]. We define $(\kappa_2, k)$-hypercontractivity as follows. This is a heavy-tailed distribution we have bound only up to the $k$-th moment.

**Definition L.1.** For integer $k \geq 4$, a distribution $P_{\mu,\Sigma}$ is $(\kappa_2, k)$-hypercontractive if for all $v \in \mathbb{R}^d$, $\mathbb{E}_{x \sim P_X}\big[|\langle v, (x - \mu)\rangle|^k\big] \leq \kappa_2^k (v^\top \Sigma v)^{k/2}$, where $\Sigma$ is the covariance.

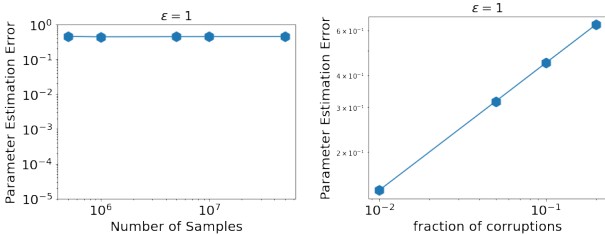

Figure 4: Performance against the stronger adversary

We give a formal description of our setting in Asmp. L.2. Note that we consider the input vector $x_i$ to be sub-Weibull and label noise $z_i$ to be hypercontractive. If both $x_i$ and $z_i$ are hypercontractive, the uncorrupted set $S_{\text{good}}$ is known to be not resilient [89, 61]. However, by [89, Lemma G.10], we can clip $x_i$ by $O(\sqrt{d}\|\Sigma\|_2)$, and obtain a $(\alpha, O(\kappa\alpha^{1-1/k}), O(\kappa\alpha^{1-2/k}), O(\kappa\alpha^{1-2/k}), O(\kappa\alpha^{1-1/k}))$-resilient set [61, Lemma 4.19]. This would result in sub-optimal error rate $\tilde{O}(\kappa\alpha^{1-2/k})$, which depends on condition number $\kappa$. For convenience, in this section, we further assume that $x_i$ and $z_i$ are independent. In the dependent case, the only thing we need to change is the $\rho_1$ resilience from $O(\alpha^{1-1/k})$ to $O(\alpha^{1-2/k})$ in Lemma L.4. This would result in $O(\alpha^{1-3/k})$ error rate if we plug this new resilience in Thm. L.5.

**Assumption L.2** (($\Sigma, \sigma^2, w^*, K, a, \kappa_2, k$)-model). A multiset $S_{\text{good}} = \{(x_i \in \mathbb{R}^d, y_i \in \mathbb{R})\}_{i=1}^n$ of $n$ i.i.d. samples is from a linear model $y_i = \langle x_i, w^* \rangle + z_i$, where the input vector $x_i$ is zero mean, $\mathbb{E}[x_i] = 0$, with a positive definite covariance $\Sigma := \mathbb{E}[x_i x_i^\top] \succ 0$, and the independent label noise $z_i$ is zero mean, $\mathbb{E}[z_i] = 0$, with variance $\sigma^2 := \mathbb{E}[z_i^2]$. We assume that the marginal distribution of $x_i$ is $(K, a)$-sub-Weibull and that of $z_i$ is $(\kappa_2, k)$-hypercontractive, as defined above.

This is similar to the light-tailed case in Asmp. H.2. The main difference is that the noise $z_i$ is heavy-tailed and independent of the input $x_i$.

**Assumption L.3** ($\alpha_{\text{corrupt}}$-corruption). Given a dataset $S_{\text{good}} = \{(x_i, y_i)\}_{i=1}^n$, an adversary inspects all the data points, selects $\alpha_{\text{corrupt}} n$ data points denoted as $S_r$, and replaces the labels with arbitrary labels while keeping the covariates unchanged. We let $S_{\text{bad}}$ denote this set of $\alpha_{\text{corrupt}} n$ newly labelled examples by the adversary. Let the resulting set be $S := S_{\text{good}} \cup S_{\text{bad}} \setminus S_r$. We further assume that the corruption rate is bounded by $\alpha_{\text{corrupt}} \leq \bar{\alpha}$, where $\bar{\alpha}$ is a positive constant that depends on $\kappa_2, k, K, \log(\kappa), a$ and $\zeta$.

Compared to Asmp. 3.7, this only difference is in the conditions on $\bar{\alpha}$. Similar as Lemma J.7, we have the following lemma showing that under Asmp. L.2, the uncorrupted dataset can $S_{\text{good}}$ is corrupt-good, which means that it can be seen as being corrupted from a resilient set. We provide the proof in App. L.2.

**Lemma L.4.** *A multiset of i.i.d. labeled samples $S_{\text{good}} = \{(x_i, y_i)\}_{i=1}^n$ is generated from a linear model: $y_i = \langle x_i, w^* \rangle + z_i$, where feature vector $x_i$ has zero mean and covariance $\mathbb{E}[x_i x_i^\top] = \Sigma \succ 0$, independent label noise $z_i$ has zero mean and covariance $\mathbb{E}[z_i^2] = \sigma^2 > 0$. Suppose $x_i$ is $(K, a)$-sub-Weibull, $z_i$ is $(\kappa_2, k)$-hypercontractive, then there exist constants $c_1, C_2 > 0$ such that, for any $0 < \alpha \leq \bar{\alpha} \leq c$ where $c \in (0, 1/2)$ is some absolute constant if*

$$n \geq c_1 \left( \frac{d}{\zeta^{2(1-1/k)}\alpha^{2(1-1/k)}} + \frac{k^2\alpha^{2-2/k}d\log d}{\zeta^{2-4/k}\kappa_2^2} + \frac{\kappa_2^2 d\log d}{\alpha^{2/k}} + \frac{d + \log(1/\zeta)}{\tilde{\alpha}^2} \right) , \qquad (87)$$

*then with probability $1 - \zeta$, $S_{\text{good}}$ is*
$(0.2\alpha, \alpha, C_2 k(ka)^a K\kappa_2\alpha^{1-1/k}\zeta^{-1/k}, C_2 K^2\tilde{\alpha}\log^{2a}(1/\tilde{\alpha}), C_2 k^2\kappa_2^2\alpha^{1-2/k}\zeta^{-2/k}, C_2 K\tilde{\alpha}\log^a(1/\tilde{\alpha}))$-*corrupt good with respect to $(w^*, \Sigma, \sigma)$.*

In the rest of this section, we assume we have a $(O(\alpha), \alpha, \rho_1, \rho_2, \rho_3, \rho_4)$-corrupt good set under Asmp. L.2 and present following algorithm and our main theorem under this setting in Thm. L.5. We also provide the proof in App. L.1.

---

**Algorithm 4:** Robust and Private Linear Regression for heavy-tailed noise

---

**Input:** dataset $S = \{(x_i, y_i)\}_{i=1}^{3n}$, $(\varepsilon, \delta)$, $T$, learning rate $\eta$, failure probability $\zeta$, target error rate $\alpha$, distribution parameter $(K, a)$

1   Partition dataset $S$ into three equal sized disjoint subsets $S = S_1 \cup S_2 \cup S_3$.

2   $\delta_0 \leftarrow \delta/(2T)$, $\varepsilon_0 \leftarrow \varepsilon/(4\sqrt{T\log(1/\delta_0)})$, $\zeta_0 \leftarrow \zeta/3$, $w_0 \leftarrow 0$

3   $\Gamma \leftarrow \text{PrivateNormEstimator}(S_1, \varepsilon_0, \delta_0, \zeta_0)$, $\Theta \leftarrow K\sqrt{2\Gamma}\log^a(n/\zeta_0)$

4   **for** $t = 1, 2, \ldots, T-1$ **do**

5      $\gamma_t \leftarrow \text{RobustPrivateDistanceEstimator}(S_2, w_t, \varepsilon_0, \delta_0, \alpha, \zeta_0)$

6      $\theta_t \leftarrow 2\sqrt{2\gamma_t} \cdot \sqrt{\max\{8\rho_2/\alpha, 8\rho_3/\alpha\} + 1}$.

7      Sample $\nu_t \sim \mathcal{N}(0, \mathbf{I}_d)$

8      $w_{t+1} \leftarrow w_t - \eta\left(\frac{1}{n}\sum_{i \in S_3}\left(\text{clip}_\Theta(x_i)\text{clip}_{\theta_t}\left(w_t^\top x_i - y_i\right)\right) + \frac{\sqrt{2\log(1.25/\delta_0)}\Theta\theta_t}{\varepsilon_0 n} \cdot \nu_t\right)$

9   Return $w_T$

---

**Theorem L.5.** *Alg. 4 is $(\varepsilon, \delta)$-DP. Under $(\Sigma, \sigma^2, w^*, K, a, \kappa_2, k)$-model of Asmp. L.2 and $\alpha_{\text{corrupt}}$-corruption of Assumption L.3 and for any failure probability $\zeta \in (0,1)$ and target error rate $\alpha \geq 1.2\alpha_{\text{corrupt}}$, if the dataset $S$ is $(0.2\alpha, \alpha, \rho_1, \rho_2, \rho_3, \rho_4)$-corrupt good set $S$ with respect to $(w^*, \Sigma, \sigma)$ and sample size is large enough such that*

$$n = O\left(K^2 d\log(d/\zeta)\log^{2a}(n/\zeta) + \frac{K^2 dT^{1/2}\log(T/\delta)\log^a(n/(\alpha\zeta))\sqrt{8\max\{\rho_2/\alpha, \rho_3/\alpha\} + 1}}{\varepsilon\hat{\rho}(\alpha)}\right),$$

(88)

*where $\hat{\rho}(\alpha) = \max\{\rho_1, 3\rho_2, 2\rho_4\sqrt{8\max\{\rho_2/\alpha, \rho_3/\alpha\} + 1}\}$, then the choices of a small enough step size, $\eta \leq 1/(1.1\lambda_{\max}(\Sigma))$, and the number of iterations, $T = \tilde{\Theta}(\kappa\log(\|w^*\|))$ for a condition number of the covariance $\kappa := \lambda_{\max}(\Sigma)/\lambda_{\min}(\Sigma)$, ensures that, with probability $1 - \zeta$, Alg. 1 achieves*

$$\mathbb{E}_{\nu_1, \cdots, \nu_t \sim \mathcal{N}(0, \mathbf{I}_d)}\left[\|w_T - w^*\|_\Sigma^2\right] = \tilde{O}\left(\hat{\rho}^2(\alpha)\sigma^2\right),$$

(89)

*where the expectation is taken over the noise added for DP, and $\tilde{\Theta}(\cdot)$ hides logarithmic terms in $K, \kappa_2, \sigma, d, n, 1/\varepsilon, \log(1/\delta), 1/\alpha$, and $\kappa$.*

By Lemma L.4, if we set $\tilde{\alpha} = \alpha^{1-1/k}$, $\rho_1 = C_2 k(ka)^a K\kappa_2\alpha^{1-1/k}\zeta^{-1/k}$, $\rho_2 = C_2 K^2\alpha^{1-1/k}\log^{2a}(1/\alpha^{1-1/k})$, $\rho_3 = C_2 k^2\kappa_2^2\alpha^{1-2/k}\zeta^{-2/k}$, and $\rho_4 = C_2 K\alpha^{1-1/k}\log^a(1/\alpha^{1-1/k})$, we have following corollary.

**Corollary L.6.** *Under the same hypotheses of Thm. L.5 and under $\alpha_{\text{corrupt}}$-corruption model of Asmp. L.3, if $1.2\alpha_{\text{corrupt}} \leq \alpha$ and $K, a, \kappa_2, k = O(1)$, then $n = \tilde{O}(d/(\zeta^{2-2/k}\alpha^{2-2/k}) + \kappa^{1/2}d\log(1/\delta)/(\varepsilon\alpha^{1-1/k}))$ samples are sufficient for Alg. 4 to achieve an error rate of $(1/\sigma^2)\|\hat{w} - w^*\|_\Sigma^2 = \tilde{O}(\zeta^{-2/k}\alpha^{2-4/k})$ with probability $1 - \zeta$, where $\kappa := \lambda_{\max}(\Sigma)/\lambda_{\min}(\Sigma)$, $\tilde{O}(\cdot)$ hides logarithmic terms in $\sigma, d, n, 1/\varepsilon, \log(1/\delta), \log(1/\zeta)$ and $\kappa$.*

Simiarly, if we set $\tilde{\alpha} = \alpha$, $\rho_1 = C_2 k(ka)^a K\kappa_2\alpha^{1-1/k}\zeta^{-1/k}$, $\rho_2 = C_2 K^2\alpha\log^{2a}(1/\alpha)$, $\rho_3 = C_2 k^2\kappa_2^2\alpha^{1-2/k}\zeta^{-2/k}$, and $\rho_4 = C_2 K\alpha\log^a(1/\alpha)$, we have following corollary.

**Corollary L.7.** *Under the same hypotheses of Thm. L.5 and under $\alpha_{\text{corrupt}}$-corruption model of Asmp. L.3, if $1.2\alpha_{\text{corrupt}} \leq \alpha$ and $K, a, \kappa_2, k = O(1)$, then $n = \tilde{O}(d/(\zeta^{2-2/k}\alpha^{2-2/k}) + \kappa^{1/2}d\log(1/\delta)/(\varepsilon\alpha) + (d + \log(1/\zeta)/\alpha^2))$ samples are sufficient for Alg. 4 to achieve an error rate of $(1/\sigma^2)\|\hat{w} - w^*\|_\Sigma^2 = \tilde{O}(\zeta^{-2/k}\alpha^{2-2/k})$ with probability $1 - \zeta$, where $\kappa := \lambda_{\max}(\Sigma)/\lambda_{\min}(\Sigma)$, $\tilde{O}(\cdot)$ hides logarithmic terms in $\sigma, d, n, 1/\varepsilon, \log(1/\delta), \log(1/\zeta)$ and $\kappa$.*

As a comparison, we also apply the exponential-time robust linear regression algorithm HPTR by [61] under our setting.

**Theorem L.8** ([61, Theorem 12]). *There exist positive constants $c$ and $C$ such that for any $((2/11)\alpha, \alpha, \rho_1, \rho_2, \rho_3, \rho_4)$-corrupt good set $S$ with respect to $(w^*, \Sigma \succ 0, \sigma > 0)$ satisfying $\alpha < c$,*

$\rho_1 < c$, $\rho_2 < c$, $\rho_3 < c$,and $\rho_4^2 \le c\alpha$, HPTR *achieves* $(1/\sigma)\|(\hat{\beta} - \beta)\|_\Sigma \le 32\rho_1$ *with probability* $1 - \zeta$, *if*

$$n \ge C \frac{d + \log(1/(\delta\zeta))}{\varepsilon\alpha} . \tag{90}$$

We set $\tilde{\alpha} = \alpha^{1-1/k}$, $\rho_1 = C_2 k(ka)^a K \kappa_2 \alpha^{1-1/k} \zeta^{-1/k}$, $\rho_2 = C_2 K^2 \alpha^{1-1/k} \log^{2a}(1/\alpha^{1-1/k})$,$\rho_3 = C_2 k^2 \kappa_2^2 \alpha^{1-2/k} \zeta^{-2/k}$, and $\rho_4 = C_2 K \alpha^{1-1/k} \log^a(1/\alpha^{1-1/k})$, we have the following utility gaurentees.

**Corollary L.9.** *Under the hypothesis of Asmp. L.2, there exists a constant $c > 0$ such that for any $\alpha \le c$, $(ka)^a K \kappa_2 \alpha^{1-1/k} \zeta^{-1/k} \le c$, $k^2 \kappa_2^2 \alpha^{1-2/k} \zeta^{-2/k} \le c$ and $K^2 \alpha^{1-2/k} \log^{2a}(1/\alpha^{1-1/k}) \le c$, it is sufficient to have a dataset of size*

$$n = O\Big( \frac{d}{\zeta^{2(1-1/k)}\alpha^{2(1-1/k)}} + \frac{k^2\alpha^{2-2/k}d\log d}{\zeta^{2-4/k}\kappa_2^2} + \frac{\kappa_2^2 d\log d}{\alpha^{2/k}} \Big) , \tag{91}$$

*such that* HPTR *achieves* $(1/\sigma)\|\hat{w} - w^*\|_\Sigma = O(k(ka)^a K \kappa_2 \alpha^{1-1/k} \zeta^{-1/k})$ *with probability* $1 - \zeta$.

Note that both of our result in Corollary L.6 and Corollary L.7 are suboptimal compared to the exponential time algorithm HPTR from Corollary L.9. Suppose $K, a, \kappa_2, k, \zeta = \Theta(1)$, HPTR achieves $(1/\sigma)\|w^* - \hat{w}\| = \tilde{O}(\alpha^{1-1/k})$ with sample complexities $n = d/(\alpha^{2(1-1/k)}) + (d + \log(1/\delta))/(\varepsilon n)$. However, in the analysis in Corollary L.6, Alg. 4 achieves $(1/\sigma)\|w^* - \hat{w}\| = \tilde{O}(\alpha^{1-2/k})$ with the same sample complexities. In the analysis in Corollary L.7, Alg. 4 achieves the same error rate as HPTR but requires extra $\tilde{O}(d/\alpha^2)$ sample complexities. The suboptimality is caused by the gradient truncation step in our algorithm. From Thm. L.8, the final error rate of HPTR only depends on the first resilience $\rho_1$. However in Thm. L.5, the final error rate of Alg. 4 depends on $\hat{\rho}(\alpha) = \max\{\rho_1, \rho_2, \rho_4\sqrt{\rho_2/\alpha}\}$. When the noise is heavy-tailed, the bottleneck is the last term $\rho_4\sqrt{\rho_2/\alpha} \approx \alpha^{1-2/k}$, which is due to the truncation threshold from Eq. (101). This cannot be tightened by using a smaller truncation threshold. Because we can construct $y_i$, such that there are $\alpha$-fraction of points that are at the threshold level $\theta_t \approx \alpha^{-1/k}$(line 6 of Alg. 4). If exponential time complexity is allowed, we could robustly and privately estimate the average of the gradients by directly estimating the $x_i y_i$. However, the current best efficient algorithm [60] for estimating the mean of Gaussian with unknown covariance robustly and privately would require $O(d^{1.5})$ samples.

For a fair comparison, we also rewrite the error rates of Corollary L.6, Corollary L.7, Corollary L.9 as the same accuracy level $\alpha$ and different corruption level $\alpha_{\text{corrupt}}$ respectively.

**Corollary L.10.** *Under the same hypotheses of Thm. L.5 and under $\alpha_{\text{corrupt}}$-corruption model of Asmp. L.3, if $1.2\alpha_{\text{corrupt}} \le \alpha^{k/(k-2)}$ and $K, a, \kappa_2, k = O(1)$, then*

$$n = \tilde{O}(d/(\zeta^{2-2/k}\alpha^{2(k-1)/(k-2)}) + \kappa^{1/2}d\log(1/\delta)/(\varepsilon\alpha^{(k-1)/(k-2)}))$$

*samples are sufficient for Alg. 4 to achieve an error rate of $(1/\sigma^2)\|\hat{w} - w^*\|_\Sigma^2 = \tilde{O}(\zeta^{-2/k}\alpha^2)$ with probability $1 - \zeta$, where $\kappa := \lambda_{\max}(\Sigma)/\lambda_{\min}(\Sigma)$, $\tilde{O}(\cdot)$ hides logarithmic terms in $\sigma, d, n, 1/\varepsilon, \log(1/\delta), \log(1/\zeta)$ and $\kappa$.*

**Corollary L.11.** *Under the same hypotheses of Thm. L.5 and under $\alpha_{\text{corrupt}}$-corruption model of Asmp. L.3, if $1.2\alpha_{\text{corrupt}} \le \alpha^{k/(k-1)}$ and $K, a, \kappa_2, k = O(1)$, then*

$$n = \tilde{O}(d/(\zeta^{2-2/k}\alpha^2) + \kappa^{1/2}d\log(1/\delta)/(\varepsilon\alpha^{k/(k-1)}) + (d + \log(1/\zeta)/\alpha^{2k/(k-1)}))$$

*samples are sufficient for Alg. 4 to achieve an error rate of $(1/\sigma^2)\|\hat{w} - w^*\|_\Sigma^2 = \tilde{O}(\zeta^{-2/k}\alpha^2)$ with probability $1 - \zeta$, where $\kappa := \lambda_{\max}(\Sigma)/\lambda_{\min}(\Sigma)$, $\tilde{O}(\cdot)$ hides logarithmic terms in $\sigma, d, n, 1/\varepsilon, \log(1/\delta), \log(1/\zeta)$ and $\kappa$.*

**Corollary L.12** (HPTR)**.** *Under the same hypotheses of Thm. L.5 and under $\alpha_{\text{corrupt}}$-corruption model of Asmp. L.3, if $\alpha_{\text{corrupt}} \le \alpha^{k/(k-1)}$ and $\alpha^{(k-2)/(k-1)} \le c$ and $K, a, \kappa_2, k = O(1)$, then*

$$n = \tilde{O}\Big( \frac{d}{\zeta^{2-2/k}\alpha^2} + \frac{d + \log(1/(\delta\zeta))}{\varepsilon\alpha^{k/k-1}} \Big)$$

*samples are sufficient for HPTR to achieve an error rate of $(1/\sigma^2)\|\hat{w} - w^*\|_\Sigma^2 = \tilde{O}(\zeta^{-2/k}\alpha^2)$ with probability $1 - \zeta$, $\tilde{O}(\cdot)$ hides logarithmic terms in $\sigma, d, n, 1/\varepsilon, \log(1/\delta), \log(1/\zeta)$ and $\kappa$.*

### L.1 Proof of Thm. L.5

*Proof.* The proof follows similarly as the proof of Thm. H.3. We only highlight the difference in the proof.

Let $S_{\text{good}}$ be the uncorrupted dataset for $S_3$ and $S_{\text{bad}}$ be the corrupted data points in $S_3$. Let $G$ denote the clean data that satisfies resilience conditions. We know $|G| \geq (1 - 1.2\alpha_{\text{corrupt}})n \geq (1 - \alpha)n$.

Let $\lambda_{\max} = \|\Sigma\|_2$. Define $\hat{\Sigma} := (1/n)\sum_{i \in G} x_i x_i^\top$, $\hat{B} := \mathbf{I}_d - \eta\hat{\Sigma}$. Lemma J.4 implies that if $n = O(K^2 d \log(d/\zeta) \log^{2a}(n/\zeta))$, then

$$0.9\Sigma \preceq \hat{\Sigma} \preceq 1.1\Sigma . \tag{92}$$

We pick step size $\eta$ such that $\eta \leq 1/(1.1\lambda_{\max})$ to ensure that $\eta \leq 1/\|\hat{\Sigma}\|_2$. Since the covariates $\{x_i\}_{i \in S}$ are not corrupted, from Lemma J.3, we know with probability $1 - \zeta$, for all $i \in S_3$,

$$\|x_i\|^2 \leq K^2 \operatorname{Tr}(\Sigma) \log^{2a}(n/\zeta) . \tag{93}$$

The rest of the proof is under Eq. (92), Eq. (93) and the resilience conditions.

Let $\phi_t = (\sqrt{2\log(1.25/\delta_0)}\Theta\theta_t)/(\varepsilon_0 n)$. Define $g_i^{(t)} := x_i(x_i^\top w_t - y_i)$. For $i \in S_{\text{good}}$, we know $y_i = x_i^\top w^* + z_i$. Let $\tilde{g}_i^{(t)} = \operatorname{clip}_\Theta(x_i)\operatorname{clip}_{\theta_t}(x_i^\top w_t - y_i)$. Note that under Eq. (93), $\operatorname{clip}_\Theta(x_i) = x_i$ for all $i \in S_3$.

From Alg. 4, we can write one-step update rule as follows:

$$w_{t+1} - w^*$$
$$= w_t - \eta\left(\frac{1}{n}\sum_{i \in S}\tilde{g}_i^{(t)} + \phi_t\nu_t\right) - w^*$$
$$= \left(\mathbf{I} - \frac{\eta}{n}\sum_{i \in G}x_i x_i^\top\right)(w_t - w^*) + \frac{\eta}{n}\sum_{i \in G}x_i z_i + \frac{\eta}{n}\sum_{i \in G}(g_i^{(t)} - \tilde{g}_i^{(t)}) - \eta\phi_t\nu_t - \frac{\eta}{n}\sum_{i \in S_3 \setminus G \cup E_t}\tilde{g}_i^{(t)} \tag{94}$$

Let $E_t := \{i \in G : \theta_t \leq |x_i^\top w_t - y_i|\}$ be the set of clipped clean data points such that $\sum_{i \in G}(g_i^{(t)} - \tilde{g}_i^{(t)}) = \sum_{i \in E_t}(g_i^{(t)} - \tilde{g}_i^{(t)})$. We define $\hat{v} := (1/n)\sum_{i \in G}x_i z_i$, $u_t^{(1)} := (1/n)\sum_{i \in E_t}x_i x_i^\top(w_t - w^*)$, $u_t^{(2)} := (1/n)\sum_{i \in E_t}-x_i z_i$, and $u_t^{(3)} := (1/n)\sum_{i \in S_3 \setminus G \cup E_t}\tilde{g}_i^{(t)}$.

We can further write the update rule as:

$$w_{t+1} - w^* = \hat{B}(w_t - w^*) + \eta\hat{v} + \eta u_{t-1}^{(1)} + \eta u_{t-1}^{(2)} - \eta\phi_t\nu_t - \eta u_{t-1}^{(3)} . \tag{95}$$

Since $G \subset S_{\text{good}}$ and $|G| \geq (1 - \alpha)n$, using the resilience property in Eq. (5), we know

$$\|\Sigma^{-1/2}\hat{v}\| = |G| \max_{\|v\|=1}\Sigma^{-1/2}\left\langle v, \frac{1}{|G|}\sum_{i \in G}x_i z_i\right\rangle$$
$$\leq (1 - \alpha)\rho_1\sigma \tag{96}$$
$$\leq \rho_1\sigma . \tag{97}$$

Let $\alpha_2 = |E_t|/n$. Following the proof of Lemma C.3, we can show following lemma.

**Lemma L.13.** *Under Assumptions L.2, if $\theta_t \geq \sqrt{\max\{8\rho_2/\alpha, 8\rho_3/\alpha\} + 1} \cdot (\|w^* - w_t\|_\Sigma + \sigma)$, then*

$$\left|\{i \in G : |w_t^\top x_i - y_i| \geq \theta_t\}\right| \leq \alpha n$$

*, for all $t \in [T]$.*

Similar as Thm. C.1, we have following theorem.

**Theorem L.14.** *Alg. 2 is $(\varepsilon_0, \delta_0)$-DP. For an $(\alpha_{\text{corrupt}}, \bar{\alpha}, \rho_1, \rho_2, \rho_3, \rho_4)$-corrupted good dataset $S_2$ and an upper bound $\bar{\alpha}$ on $\alpha_{\text{corrupt}}$ that satisfy Asmp. L.2 and $\rho_1 + \rho_2 + \rho_3 \leq 1/4$, for any $\zeta \in (0,1)$, if*

$$n = O\left(\frac{\log(1/\zeta)\log(1/(\delta_0\zeta))}{\bar{\alpha}\varepsilon_0}\right), \tag{98}$$

*with a large enough constant then, with probability $1 - \zeta$, Alg. 2 returns $\ell$ such that $\frac{1}{4}(\|w_t - w^*\|_\Sigma^2 + \sigma^2) \leq \ell \leq 4(\|w_t - w^*\|_\Sigma^2 + \sigma^2)$.*

This means $\alpha_2 \leq \alpha$, and we have

$$\|\Sigma^{-1/2}u_t^{(1)}\| = \|\Sigma^{-1/2}\frac{1}{n}\sum_{i \in E_t} x_i x_i^\top (w_t - w^*)\| .$$

From Corollary J.8, we know

$$\left| \|\Sigma^{-1/2}\frac{1}{|E_t|}\sum_{i \in E_t} x_i x_i^\top (w_t - w^*)\| - \|w_t - w^*\|_\Sigma \right|$$

$$= \left| \max_{u:\|u\|=1} \frac{1}{|E_t|}\sum_{i \in E_t} u^\top \Sigma^{-1/2} x_i x_i^\top (w_t - w^*)\| - \max_{v:\|v\|=1} v^\top \Sigma^{1/2}(w_t - w^*) \right|$$

$$\leq \max_{u:\|u\|=1} \left| \frac{1}{|E_t|}\sum_{i \in E_t} u^\top \Sigma^{-1/2} x_i x_i^\top \Sigma^{-1/2}\Sigma^{1/2}(w_t - w^*)\| - u^\top \Sigma^{1/2}(w_t - w^*) \right|$$

$$\leq \max_{u:\|u\|=1} \left| \frac{1}{|E_t|}\sum_{i \in E_t} u^\top \left(\Sigma^{-1/2} x_i x_i^\top \Sigma^{-1/2} - \mathbf{I}_d\right)\Sigma^{1/2}(w_t - w^*)\| \right|$$

$$= \left\| \frac{1}{|E_t|}\sum_{i \in E_t} \left(\Sigma^{-1/2} x_i x_i^\top \Sigma^{-1/2} - \mathbf{I}_d\right)\Sigma^{1/2}(w_t - w^*) \right\|$$

$$\leq \left\| \frac{1}{|E_t|}\sum_{i \in E_t} \left(\Sigma^{-1/2} x_i x_i^\top \Sigma^{-1/2} - \mathbf{I}_d\right) \right\| \cdot \left\| \Sigma^{1/2}(w_t - w^*) \right\|$$

$$\leq \frac{2 - \alpha_2}{\alpha_2}\rho_2 \|w_t - w^*\|_\Sigma .$$

This implies that

$$\|\Sigma^{-1/2}u_t^{(1)}\| \leq \|\Sigma^{-1/2}\frac{1}{n}\sum_{i \in E} x_i x_i^\top (w_t - w^*)\|$$
$$\leq (\alpha_2 + 2\rho_2)\|w_t - w^*\|_\Sigma$$
$$\leq 3\rho_2 \|w_t - w^*\|_\Sigma , \tag{99}$$

where the last inequality follows from the fact that $\alpha_2 \leq \alpha$ and our assumption that $\alpha \leq \rho_2$ from Asmp. L.3. Similarly, we use resilience property in Eq. (5) instead of Eq. (6), we can show that

$$\|\Sigma^{-1/2}u_t^{(2)}\| \leq 3\rho_3\sigma . \tag{100}$$

Next, we consider $u_t^{(3)}$. Since $|S_3 \setminus G| \leq 1.2\alpha_{\text{corrupt}}n$ and $|E_t| \leq \alpha n$, using Eq. (8) and Corollary J.8, we have

$$\|\Sigma^{-1/2}u_t^{(3)}\| = \max_{v:\|v\|=1} \frac{1}{n}\sum_{i \in S_{\text{bad}} \cup E_t} v^\top \Sigma^{-1/2} x_i \text{clip}_{\theta_t}(x_i^\top w_t - y_i)$$

$$\leq 2\rho_4\theta_t$$

$$\leq 2\rho_4\sqrt{8\max\{\rho_2/\alpha, \rho_3/\alpha\} + 1} \cdot (\|w_t - w^*\|_\Sigma + \sigma) . \tag{101}$$

The analysis of convergence follows similarly as in Step 3 and Step 4 of the proof of Thm. H.3 except we set $\hat{\rho}(\alpha) = \max\{\rho_1, 3\rho_2, 2\rho_4\sqrt{8\max\{\rho_2/\alpha, \rho_3/\alpha\} + 1}\}$.

The second term in Eq. (88) ensures the added Gaussian noise is small enough such that $\phi_t^2\|v_t\|^2 \leq \hat{\rho}(\alpha)^2(\mathbb{E}[\|w_t - w^*\|_\Sigma^2] + \sigma^2)$, which is similar as in Eq. (59)

$\square$

## L.2  Proof of Lemma L.4

*Proof.* For any $x$ that is $(K, a)$-sub-Weibull from Definition H.2, Eq. (76) implies that for any $k \geq 1$,

$$\mathbb{E}[|\langle v, x\rangle|^k] = \int_0^\infty \mathbb{P}(|\langle v, x\rangle| \geq t^{1/k})dt \tag{102}$$

$$\leq \int_0^\infty 2\exp\left(-\frac{t^{\frac{1}{ka}}}{(K^2\mathbb{E}[\langle v, x\rangle^2])^{\frac{1}{2a}}}\right)dt \tag{103}$$

$$= 2K^k(\mathbb{E}[\langle v, x\rangle^2])^{k/2}ka\int_0^\infty e^{-u}u^{ka-1}du \tag{104}$$

$$= 2K^k(\mathbb{E}[\langle v, x\rangle^2])^{k/2}\Gamma(ka + 1) \tag{105}$$

$$\leq 2K^k(\mathbb{E}[\langle v, x\rangle^2])^{k/2}(ka)^{ka} \tag{106}$$

This implies that $x_i$ is also $((ka)^a K, k)$-hypercontractive. Since $x_i$ and $z_i$ are independent, we have

$$\mathbb{E}\left[\left|\langle v, \sigma^{-1}\Sigma^{-1/2}x_iz_i\rangle\right|^k\right] = \mathbb{E}\left[\left|\langle v, \Sigma^{-1/2}x_i\rangle\right|^k\right]\mathbb{E}\left[|\sigma^{-1}z_i|^k\right] \leq 2(ka)^{ka}K^k\kappa_2^k . \tag{107}$$

This means $x_iz_i$ is also $((ka)^a K\kappa_2, k)$-hypercontractive. From [89, Lemma G.10], we know with probability $1 - \zeta$, there exists $S_1 \subset S_{\text{good}}$ with $|S_1| \geq (1 - 0.1\alpha)|S_{\text{good}}|$, such that for any $T \subset S_1$ with $|T| \geq (1 - \alpha)|S_1|$, we have

$$\left|\frac{1}{|T|}\sum_{(x_i,y_i)\in S}\langle v, \sigma^{-1}\Sigma^{-1/2}x_i(y_i - x_i^\top w^*)\rangle\right| \leq C_2k(ka)^a K\kappa_2\alpha^{1-1/k}\zeta^{-1/k} . \tag{108}$$

Similarly, there exists $S_2 \subset S_{\text{good}}$ with $|S_2| \geq (1 - 0.1\alpha)|S_{\text{good}}|$, such that for any $T \subset S_2$ with $|T| \geq (1 - \alpha)|S_2|$, we have

$$\left|\frac{1}{|T|}\sum_{(x_i,y_i)\in T}(\sigma^{-1}(y_i - x_i^\top w^*))^2 - 1\right| \leq C_2k^2\kappa_2^2\alpha^{1-2/k}\zeta^{-2/k} . \tag{109}$$

From Lemma J.7, for any $T \subset S_{\text{good}}$ with $|T| \geq (1 - \tilde{\alpha})|S_{\text{good}}|$, we have

$$\left|\frac{1}{|T|}\sum_{(x_i,y_i)\in T}\langle v, \Sigma^{-1/2}x_i\rangle^2 - 1\right| \leq C_2K\tilde{\alpha}\log^{2a}(1/\tilde{\alpha}) . \tag{110}$$

and

$$\left|\frac{1}{|T|}\sum_{(x_i,y_i)\in T}\langle v, \Sigma^{-1/2}x_i\rangle\right| \leq C_2K\tilde{\alpha}\log^a(1/\tilde{\alpha}) . \tag{111}$$

Set $S = S_1 \cap S_2$, we know $|S| \geq (1 - 0.2\alpha)|S_{\text{good}}|$ and $S$ is
$(0.2\alpha, \alpha, C_2k(ka)^a K\kappa_2\alpha^{1-1/k}\zeta^{-1/k}, C_2K^2\tilde{\alpha}\log^{2a}(1/\tilde{\alpha}), C_2k^2\kappa_2^2\alpha^{1-2/k}\zeta^{-2/k}, C_2K\tilde{\alpha}\log^a(1/\tilde{\alpha}))$-corrupt good with respect to $(w^*, \Sigma, \sigma)$. This completes the proof. $\square$