# OpenReview forum: "Label Robust and Differentially Private Linear Regression: Computational and Statistical Efficiency"
_NeurIPS.cc/2023/Conference — NeurIPS 2023 poster_

### Official Review · Reviewer_yhsn · 2023-07-03

**Soundness:** 3 good
**Presentation:** 3 good
**Contribution:** 2 fair
**Rating:** 5
**Confidence:** 2

**Summary:**

This work studies the problem of $(\varepsilon, \delta)$-differently private linear regression. The covariates are drawn from a Gaussian distribution $N(0,\Sigma)$ with unknown covariance $\Sigma$ and the random noise in the model is either Gaussian or heavier-tailed. Further, a small fraction of the labels can be corrupted. The error is measured as $\lVert \Sigma^{1/2} (\hat{w} - w^*) \rVert_2$ (when the variance of the random noise equals 1), where $w^*$ is the true regression vector.

Let $\kappa$ be the condition number of $\Sigma$. For the case that the noise is Gaussian the authors propose an algorithm that achieves error $\alpha$ in polynomial time using $\tilde{O}(\tfrac d {\alpha^2} + \sqrt{\kappa} \cdot \tfrac d {\varepsilon \alpha})$ samples. Previous efficient algorithms for this task needed $\tilde{O}(\tfrac d {\alpha^2} + \kappa \cdot \tfrac d {\varepsilon \alpha} + \kappa^2 \tfrac d \varepsilon)$. On the other hand, it is known that  $\Omega(\tfrac d {\alpha^2} + \tfrac d {\varepsilon \alpha})$ samples are necessary information-theoretically. This can be achieved by inefficient algorithms, i.e., there is no dependence on the condition number.

Their results for heavy-tailed noise are similar to the results for Gaussian noise. Previously there were no known results in this setting. Their algorithm is based on differentially private stochastic gradient descent (DP-SGD).

**Strengths:**

Previous works have also used DP-SGD for this problem. The authors give a more careful analysis (and tweak it in some places) which yields an algorithm with an improved sample complexity. One of their tweaks allows them to also handle corruptions in the labels. It is nice to see that this algorithmic approach can be made robust to corruptions in the labels.

Overall, the paper is well-written.

**Weaknesses:**

Qualitatively, there is still a dependence on the condition number. So this work does not give new insights into whether this is necessary or not.

**Questions:**

Since both you and previous work analyze DP-SGD and incur (some) dependence on the condition number, is there any reason to believe that this approach might inherently have at least some dependence on this?

**Limitations:**

The authors mention open questions that are note answered by their work in multiple places.

---

> ### Author Rebuttal · Authors · 2023-08-10
>
> We thank the reviewer for the constructive feedback and address each comment as follows.
>
> 1. Polynomial dependence on $\kappa$: We kindly direct reviewer to our general response for further details.
> &nbsp;\
> &nbsp;
>
> 2. The GD-type of methods has inherent $\kappa$ dependence: Our work and the prior work [78] are the only two methods that achieve $\tilde{O}(d)$ samples for DP linear regression, even without considering robustness. The $\kappa^{0.5}$ dependence in our work arises from the compositions of $\tilde{O}(\kappa)$ iterations. Given that both algorithms are iterative in nature, it leads us to conjecture that as long as the number of iterations has a dependence on $\kappa$, complete removal of the dependence may not be feasible. One possible approach to reduce the number of iterations is applying Newton's method, which could potentially bring it down to $\tilde{O}(1)$. However, it remains unclear how to privatize the OLS solution without increasing the sample complexity to $\tilde{O}(d^{1.5})$ as we need to compute $\Sigma^{-1}$ privately. As a result, while we continue to explore avenues for improvement, we believe our current work represents a significant advancement in our understanding of iterative algorithms for private linear regression. We believe that the technical advances we make in this paper, using resilience to handle the dependence in the iterates, could potentially be an important tool in understanding other private iterative algorithms, even though the (small) dependence on $\kappa$ still remains and is open if this is necessary.

---

> > ### Comment · Reviewer_yhsn · 2023-08-14
> >
> > Thank you for your detailed response. I believe it would be important to add such a discussion to the submission.

---

> > > ### Author Response · Authors · 2023-08-14
> > > **Thanks for the comments.**
> > >
> > > We will add this discussion in the revision of the paper.

---

### Official Review · Reviewer_kzNP · 2023-07-06

**Soundness:** 3 good
**Presentation:** 3 good
**Contribution:** 3 good
**Rating:** 6
**Confidence:** 2

**Summary:**

This paper introduces the first computationally and statistically efficient algorithm for differentially private and label-corruption robust linear regression. In particular, they improve over the best sample complexity achieved by non-robust algorithms in terms of the condition number of the covariance matrix. This is achieved through a variant of DP-SGD with a new adaptive clipping technique. The authors also show the empirical improvement of their proposed method over existing techniques on a synthetic dataset.

**Strengths:**

* Technically solid paper on an important topic. Some of the analysis methods are original and interesting.
* The paper is well written and the contributions are clear.

**Weaknesses:**

### Weaknesses:

* It is clear that there is an improvement in terms of sample complexity, but my main concern is the significance of the improvement, especially that a polynomial dependence on the condition number remains, which may be extremely large. Do you think it is possible to suppress the dependence on the condition number, and how?

* Is there any novelty in the proof of the lower bound in Proposition 3.9 compared to [11]?

* Can you show the exact runtimes of your algorithm compared to that of [78]?

* While I understand that the paper is mainly theoretical, I fail to think of any practical scenarios where an adversary can only corrupt labels.



### Minor comments:

* Typos: line 178: proportional -> proportionally, line 363: increase -> increases
* Equation 3: $\nu_t$ -> $\nu_T$
* Definition 2.2: (suggestion) random variables are typically denoted with capital letters
* Lemma B.2: $\infty$ -> {$\infty$}

**Questions:**

Refer to Weaknesses section for questions.


**Limitations:**

Yes.

---

> ### Author Rebuttal · Authors · 2023-08-10
>
> We thank the reviewer for the constructive feedback and address each comment as follows.
> 1. Polynomial dependence on $\kappa$: We kindly direct reviewer to our general response for further details.
> &nbsp;\
> &nbsp;
> 2. Proposition 3.9: Compared to [11], our proof is a simpler version that only covers Gaussian data. We agree with the reviewer that the technique is not particularly novel and we add the proof only for completeness.
> &nbsp;\
> &nbsp;
> 3. Exact runtime compared to [78]:  We want to emphasize that we did not try to optimize the implementation for computational efficiency and the numbers we are reporting could potentially be significantly improved. For $n=10^7$, our algorithm takes 280 seconds for each trial, and [78] takes 13 seconds for each trial. Since [78] does not provide any experiments, we implement both algorithms.
> &nbsp;\
> &nbsp;\
> In total, we process $Tn = 3\kappa \log(n)n$ samples (full-batch gradient descent for $T$ iterations) and [78] processes $n$ samples (since it is a streaming algorithm). That is why our algorithm is $3\kappa\log(n)\approx 20$ times slower. Currently, we only use a single CPU for our experiments. This gap can potentially be made significantly smaller if we parallelize the computation by using multiple GPUs. Since all of these gradients computation and distance estimation can be fully parallelized.
> &nbsp;\
> &nbsp;
>
> 4. Practical scenarios for label corruption: We thank the reviewer for pushing us to think more carefully about the motivation and will add the following discussion in the revision.
> &nbsp;\
> &nbsp;\
> First concrete scenario where only the label can be corrupted by adversaries is crowdsourced annotation. An important procedure in the standard machine learning pipeline is annotating the data using crowdsourcing platforms. For example, ImageNet, one of the most popular computer vision datasets, contains more than 14 million images hand-annotated on Amazon’s Mechanical Turk. In this case, the images are clean since the data curator had full control over the quality. However, a subset of the workers can collude to corrupt the labels of a fraction of the training data. The fraction $\alpha$ of the labels being corrupted corresponds to how much resource the adversary has to “bribe” many workers. We believe this scenario is practical, realistic, and consistent with our assumptions.
> &nbsp;\
> &nbsp;\
> Another concrete scenario which is slightly less relevant is knowledge distillation. In knowledge distillation, a trainer is given a teacher model (possibly trained by an adversary) and wants to train a student model. The trainer has clean images that are quality-controlled, but the labels are coming from the teacher model that is potentially corrupted. Although this scenario is practical, realistic, and consistent without assumptions, it is not clear in this case what $\alpha$ means since the teacher model can corrupt an arbitrary number of labels without additional cost. Perhaps the adversary has  another goal, such as not being detected by the student trainer, that limits how much the labels can be corrupted.
> &nbsp;\
> &nbsp;
>
>
>
> 5. Typos: We thank the reviewer for pointing out the typos. We will fix all the typos in the revision.

---

> > ### Comment · Reviewer_kzNP · 2023-08-15
> >
> > Thanks for the clarifications. Please include this discussion in the revision.

---

> > > ### Author Response · Authors · 2023-08-15
> > > **Thanks for the response.**
> > >
> > > We will add the discussions in the revision of the paper.

---

### Official Review · Reviewer_xavj · 2023-07-06

**Soundness:** 4 excellent
**Presentation:** 4 excellent
**Contribution:** 3 good
**Rating:** 6
**Confidence:** 3

**Summary:**

This paper focuses on differentially private linear regression with potential label corruption. The proposed algorithm is based on DP-SGD and introduces new components such as full batch-size GD and adaptive norm clipping. The paper achieves the best sample complexity under the assumption of sub-Gaussian data distribution in the absence of corruption, with a polynomial running time. Additionally, it presents the first algorithm for DP linear regression in the presence of label corruption, supported by experimental validation.

**Strengths:**

The paper is well-structured and easy to follow. Privacy and robustness are crucial in current machine learning research, making the studied topic highly relevant. The analysis of adaptive norm clipping introduces a novel technique that could have implications for future research. The results obtained in the paper are promising, and the proposed algorithm makes a substantial contribution to the field.

**Weaknesses:**

The paper's focus on a specific distribution requirement for the samples and only considering corruption in the labels limits its generalizability. It would be beneficial to explore more general cases.

**Questions:**

- Line 181: Should it be "$\Theta(\theta_t)$" instead?
- In Algorithm 1, why are the three subsets equally divided? Is it possible to improve performance by adjusting their sizes?

**Limitations:**

Yes, limitations are discussed.

---

> ### Author Rebuttal · Authors · 2023-08-10
>
> We thank the reviewer for the constructive feedback and address each comment as follows.
> 1. Distributional requirement: We thank the reviewer for pointing this out, since this was an important contribution of this paper that was not clearly written in the paper. We focused on the subGaussian case in the main text to make it easy to follow. However, our results include general distributional assumptions: the sub-Gaussian in Assumption 2.1, the sub-Weibull in Assumption I.1 and heavy-tailed in Assumption M.2. **These are not restrictive, but rather can capture a wide range of distributions we might encounter in practice. For any given distribution family, we can apply the appropriate distributional assumption from the above and apply our techniques.  This captures the correct dependence of the error rate on the tail behavior of the distribution.** For example, the most relaxed assumption is the 4-th moment bounded distribution, which any distribution for linear regression needs to satisfy and hence includes almost all distributions of interest. The cost of being inclusive in this case is the worse sample dependence of the resulting error.
> &nbsp;\
> &nbsp;\
> Firstly, between the sub-Weibull and Heavy-tailed, we cover (how our algorithm depends on) most of the known distributions. sub-Weibull is a general assumption that covers a wide range of light-tailed distributions, e.g., sub-Gaussian distribution (a=0.5), sub-exponential distribution (a=1), Gaussian distribution (K=4, a=0.5). Here, a smaller $a$ means a lighter tail and our error and sample complexity characterizes the dependence on the tail. Similarly, for heavy tailed distributions, we cover $k$-th moment bounded distributions and here larger $k$ implies lighter tail. Our error rate and sample complexity captures the dependence on this tail behavior.
> &nbsp;\
> &nbsp;\
> Secondly, these assumptions are not specific to our algorithm, as we show with counter examples that have significantly larger errors in Section 3.4 “Necessity of our assumptions” (Line 273-278.). When there is no such tail assumption, it is not possible to achieve the optimal rate with $O(d)$ samples. Fundamentally, error in private linear regression depends on the tail of the distribution regardless of the algorithm used. To capture this fundamental dependence as tightly as we can, we are making our distributional assumptions.
> &nbsp;\
> &nbsp;\
> We believe that these are not restrictive as we cover most distributions, including real world distributions (as we are only characterizing the tail and we cover almost the entire range of lightness/heaviness of the tail). We will add this discussion to the main text.
> &nbsp;\
> &nbsp;
> 2. Label corruptions: Our focus on label corruption in DP linear regression represents a crucial problem at the intersection of robustness and privacy. This setting stands out as the only one where we can achieve a sample complexity of $\tilde{O}(d)$ with computationally efficient algorithms under sub-Gaussian data. In contrast, existing algorithms dealing with adversarial corruption of both covariates and labels ([10] and [58]), are limited to exponential time complexity. Our paper identifies a non-trivial setting where near-optimality is attainable with a computationally efficient algorithm. We firmly believe that offering a near-optimal algorithm for such a case constitutes a significant contribution to the literature, especially at the intersection of robustness and privacy. It remains an open problem if a computationally efficient algorithm exists that can achieve both privacy and robustness for more general and powerful adversaries. We believe this is outside the scope of our paper and an exciting research direction.
> &nbsp;\
> &nbsp;
>
> 3. Equal sized subsets: We thank the reviewer for point this out. In Algorithm 1, for notational convenience we let $|S_1|=|S_2|=|S_3|$ and merge all sufficient conditions on $S_1$, $S_2$, and $S_3$ together. We assume that the dataset of size $3n$ is partitioned into three equal-sized sets $S_1,S_2,S_3$, each of size $n$. This choice makes the notations much simpler for presentation, while not changing the final guarantee as we do not keep track of the constant factors in sample complexity. It is true that the sufficient conditions on the sizes of $S_1, S_2$, and $S_3$ are different, and one could potentially gain in the constant factor in sample complexity from better partitioning of the dataset. In particular, it is sufficient to have $|S_1|=\tilde O(\log(1/\delta)/\varepsilon)$, $|S_2|=\tilde O(d/\varepsilon)$ and $|S_3|=\tilde O(d/\alpha^2+\kappa^{1/2}d\log(1/\delta)/(\varepsilon \alpha))$, with large enough constants. We will add this note to the revised version.
> &nbsp;\
> &nbsp;
>
> 4. Typos: We thank the reviewer for pointing out the typo. We will fix all the typos in the revision.

---

> > ### Comment · Reviewer_xavj · 2023-08-18
> >
> > Thanks for the response. I am satisfied with the reply and I would like to keep my positive ratings.

---

### Official Review · Reviewer_1PSa · 2023-07-07

**Soundness:** 3 good
**Presentation:** 3 good
**Contribution:** 2 fair
**Rating:** 5
**Confidence:** 3

**Summary:**

This paper presents a variant of the Differentially Private Stochastic Gradient Descent (DP-SGD) algorithm aimed at achieving efficient linear regression under differential privacy, even in scenarios where a portion of response variables is adversarially corrupted. The introduced methodology involves a full-batch gradient descent to optimize sample complexity and an adaptive clipping mechanism for robustness. The algorithm, exhibiting linear time complexity with respect to input size, also matches the theoretically optimal sample complexity, considering a distribution-dependent condition number factor.

**Strengths:**

The paper presents a focused yet intriguing investigation into the Label Robust and Differentially Private Linear Regression problem.

The proposed algorithm demonstrates improved sample complexity relative to existing polynomial-time algorithms.

**Weaknesses:**

The presentation of experimental results appears overly simplistic, resulting in some difficulties in interpretation. The study would greatly benefit from a more comprehensive representation of data, rather than relying solely on a single figure (Figure 1). Additionally, only one value of $\kappa$ is used.

**Questions:**

The authors assert that their methodology represents "the first efficient approach to achieve robustness and (ε, δ)-Differential Privacy (DP) simultaneously". However, previous research, as indicated in references [1] and [2], appears to have discussed the interplay between DP and robustness. Could the authors clarify the novel aspects of their approach in the context of these prior works, if any?

The manuscript refers to a variant of Stochastic Gradient Descent (SGD) being utilized in the process. Is this variation generalizable, or is it specifically tailored for the application to Linear Regression models?


[1] Phan, H., Thai, M. T., Hu, H., Jin, R., Sun, T., & Dou, D. (2020, November). Scalable differential privacy with certified robustness in adversarial learning. In International Conference on Machine Learning (pp. 7683-7694). PMLR.
[2] Lecuyer, M., Atlidakis, V., Geambasu, R., Hsu, D., & Jana, S. (2019, May). Certified robustness to adversarial examples with differential privacy. In 2019 IEEE symposium on security and privacy (SP) (pp. 656-672). IEEE.

**Limitations:**

Experimental results are overly simplistic.

---

> ### Author Rebuttal · Authors · 2023-08-10
>
> We thank the reviewer for the constructive feedback and address each comment as follows.
>
> 1. Experiments: A primary motivation for this project is to understand theoretically the fundamental tradeoffs involved in privacy, accuracy, and robustness in solving linear regression. Our algorithmic contributions can have implications for practical deep learning scenarios as we discuss below, but extensive experimental study would be outside the scope of this theoretical paper. We appreciate the reviewer’s constructive feedback but would like to respectively claim in the following that we attempted to include many experiments illustrating the effectiveness of our proposed algorithm. Specifically, in the main text, we have provided three plots in Figure 1, showcasing our algorithm's performance under non-robust settings with various choices. The first two plots utilize $\kappa=1$ mainly to showcase the dependence on other parameters, namely, the number of samples and the noise $\sigma^2$. However, this choice of $\kappa$ is not cherry-picked, as all competing algorithms suffer similarly with increasing $\kappa$. To demonstrate this fact, we have included the third plot (the rightmost panel), which illustrates that the relative performance of the algorithms remains similar as we increase the condition number $\kappa$ from 1 to 10^2. We consistently observe improvement over the best existing algorithm [78] across different choices of $\kappa$. Furthermore, in Appendix J, we have included additional experiments with strong label corruptions. As a theoretical paper, we believe we have provided more than enough details and interpretations for experiments, especially considering that prior works on DP linear regression, such as [10, 58, 78], lack any experimental evaluation.
>
> 2. Comparisons to certified robustness: In certified robustness to adversarial examples, the adversary perturbs a sample to confuse the classifier at inference time. It is a fundamentally different concept from the robustness to adversarial corruption at train time we discussed in our manuscript. Specifically, in certified robustness, the adversary is only allowed to perturb a sample up to some **norm constraints** and also **at inference time**. In contrast, our corruption model (Assumption 3.7) allows the adversary to inspect the training dataset and pick $\alpha$-fraction of **training data points** to perturb them **arbitrarily**. This constitutes a much stronger threat model. Applying methods from certified robustness to the corruption model we considered here will result in failure. Hence, we do not consider them as prior work.\
> We have provided a thorough related work study in Appendix A. Note that as of today, the only provably robust and DP linear regression papers are [10, 58], but they require exponential time. We are the first efficient algorithm to achieve label robustness and DP simultaneously.
>
> 3. Generalization to other models: Our method is specifically designed for DP and label-robust linear regression for sub-Gaussian data, and our main contribution lies in providing theoretical guarantees. However, with some light modifications, the proposed technique can potentially be applied to general deep models. Let us explain how one could make advances toward such an ambitious goal, as the reviewer suggested. Our algorithm involves adaptive clippings separately for the residual and features, which can also be applied to general models. For example, let us focus on deep neural networks trained on quadratic loss to keep notations simpler. The empirical risk we minimize is given by $\sum_{i=1}^n \frac{1}{2} (f_w(x_i)-y_i)^2$. Taking the gradient with respect to the neural network weights $w$, we obtain the gradient $G=\sum_{i=1}^n (f_w(x_i)-y_i) \nabla f_w(x_i)$. This gradient can be broken down into two terms: the residual $(f_w(x_i)-y_i)$ and the "feature" $\nabla f_w(x_i)$. Applying the intuition from linear regression, we propose clipping the residual and the feature separately to improve robustness against label corruption. \
> We strongly believe that separately clipping the residual and feature (as opposed to clipping the gradient as a whole) will enhance robustness against label corruption for deep neural networks. Empirical evidence supporting this conjecture can be found in the case of CLIP models, multimodal foundation models achieving SOTA performances in vision tasks trained on image-text pairs [R1]. As the data is scraped from the web, a significant portion of the text can be considered garbage (analogous to corrupted labels). To combat this label corruption, one standard method is filtering: (1) train a CLIP model on the corrupted data; (2) filter all the training data using the "Residual" on this trained CLIP model and discard large residuals; (3) retrain a CLIP model on the filtered dataset. This approach, first proposed by the LAION dataset, filtered using this technique [R2], improves the model's accuracy on the ImageNet benchmark by a remarkable 12% on the zero-shot classification that CLIP models are trained to do. Given the empirical success of this filter-once approach, we believe that filtering based on the residual for training deep neural networks is a very promising direction, especially when the data is noisy or adversarially corrupted. Our method applies similar filtering techniques within the training iterations, aiming to achieve similar gains by clipping the contribution of noisy data. This represents an exciting direction for training foundation models from noisy web-scale data with limited quality control.
>
> [R1] “Learning Transferable Visual Models From Natural Language Supervision”, Alec Radford et al. Proceedings of the 38th International Conference on Machine Learning, PMLR 139:8748-8763, 2021.
>
> [R2] “LAION-5B: An open large-scale dataset for training next generation image-text models”, Christoph Schuhmann et al.  NeurIPS 2022 Track Datasets and Benchmarks

---

> > ### Comment · Area_Chair_WiwR · 2023-08-18
> >
> > Dear Reviewer 1PSa,
> >
> > We would appreciate it if you could acknowledge the author's rebuttal and see if they have addressed your concerns. Thank you!
> >
> > AC

---

> > ### Comment · Reviewer_1PSa · 2023-08-18
> >
> > Thank you for your response. Despite the relatively limited scope of the setting, this paper still offers a valuable contribution to the field of Differential Privacy, given the extensive applicability of linear regression. I have adjusted my score to 5 accordingly.

---

### Author Rebuttal · Authors · 2023-08-10

We address common concerns regarding the significance of our $\kappa$ improvement. For linear regression without adversarial corruption or privacy, the dependence on $\kappa$ can be removed by simply applying an OLS solution. For linear regression with robustness and privacy simultaneously, there are exponential-time approaches that achieve optimal sample complexity without $\kappa$ dependencies [10, 58]. However, this becomes significantly more challenging for efficient algorithms even when either one of robustness or privacy is required. We are making progress in reducing the dependence on $\kappa$ for a problem that achieves both label-robustness and privacy, which we believe is significant progress in this intersection. We provide some evidence below.

As an example of how dependence on $\kappa$ can become significantly more challenging, let us consider a stronger adversarial corruption setting where the input can also be corrupted. In this case, the best known efficient result for linear regression is in [R1], which achieves sample complexity of $n=O(\kappa^2d/\alpha)$ or $n=O((d^2+\kappa d)/\alpha)$. Getting this took several important innovations over many works. It remains an open question if this $\kappa$ dependence can be removed or not. We believe our work falls under a similar effort for private and label-robust linear regression.

As another example of how dependence on $\kappa$ can become significantly more challenging, let us consider a simpler problem of private mean estimation. Over the past few years, there has been substantial effort in estimating Gaussian mean [R2, R3]. However, these methods are either exponential-time or sub-optimal in $d$ and $\kappa$. A recent breakthrough by [R4, R5] achieved efficiency and full optimality for Gaussian mean estimation with general covariance. It's worth noting that this breakthrough was a consequence of numerous papers on mean estimation, each incrementally improving the dependence on the condition number (among other things). Every one of these papers has contributed significantly to jointly enhancing our understanding of the problem and devising new techniques (both algorithmic and analytical), ultimately culminating in [R4, R5]. For linear regression, achieving complete removal of $\kappa$ dependence with an efficient algorithm remains an ongoing pursuit. While we strongly believe such progress will happen in the near future, allowing us to entirely omit the dependence on $\kappa$ with an efficient algorithm, to the best of our knowledge, no such solution currently exists. One might think that we can simply use (i) covariance-aware mean estimation from [R4,R5] to aggregate the gradients in linear regression, or (ii) apply stable covariance to least-squares and add noise. Neither of them strictly improves upon our result. The first one still suffers from number of iterations scaling as $\kappa$, where the composition of privacy costs us a factor of $\kappa^{0.5}$, which cannot be recovered; even together with all the advances we make in our paper, this costs at least $\kappa^{0.5}$ in the second error term. Without the advances in our paper, this will cost $\kappa$ factor in the second error term like [78]. The second approach of least squares does not achieve zero error even when the noise in the data (i.e. $\sigma^2$) is zero, which is strictly worse than our approach (among other issues that we will not go into details here). Nonetheless, we recognize that progress in research unfolds over time, and our submission provides several valuable tools (both algorithmic and analytical) that will play critical roles in advancing toward this goal.



[R1] “Robust regression revisited: Acceleration and improved estimation rates” Arun Jambulapati, Jerry Li, Tselil Schramm, Kevin Tian, NeurIPS 2021

[R2] “A Private and Computationally-Efficient Estimator for Unbounded Gaussians”, Gautam Kamath, Argyris Mouzakis, Vikrant Singhal, Thomas Steinke, Jonathan Ullman Proceedings of Thirty Fifth Conference on Learning Theory, PMLR 178:544-572, 2022.

[R3] “Covariance-Aware Private Mean Estimation Without Private Covariance Estimation”, Gavin Brown, Marco Gaboardi, Adam Smith, Jonathan Ullman, Lydia Zakynthinou, NeurIPS 2021

[R4] “Fast, Sample-Efficient, Affine-Invariant Private Mean and Covariance Estimation for Subgaussian Distributions”, Gavin Brown, Samuel B. Hopkins, Adam Smith, COLT 2023

[R5] “A Fast Algorithm for Adaptive Private Mean Estimation”, John Duchi, Saminul Haque, Rohith Kuditipudi, COLT 2023

---

### Decision · Program_Chairs · 2023-09-21

**Decision:**

Accept (poster)

**Comment:**

This paper studies the problem of differentially private linear regression with label robustness. The proposed algorithm seems to be novel and the theoretical results are strong.

The reviewers are positive and agree that the current paper has a solid contribution to the DP linear regression literature, especially to the setting with label corruptions. There are still concerns regarding the limited scope of the setting (only label corruptions) and experiments (too simple). I agree it is important to consider more general settings and conduct thorough empirical evaluations. But I do not think these are major concerns.

In light of this, I recommend acceptance and encourage the authors to incorporate the suggested discussions into the revision.